# The ATG8 E3-like ligases sense lysosomal damage and initiate ESCRT-mediated membrane repair

Dale P Corkery [1,2]✉, Deerada Wijayatunga[1], Benedita K L Feron [1,2], Laura K Herzog [1,2], Anastasia Knyazeva [1,2] & Yao-Wen Wu [1,2]✉

## Abstract

**After damage from pathogenic, chemical or physical stress, endo-lysosomal membranes are repaired and resealed by the endosomal sorting complex required for transport (ESCRT) machinery, but how this membrane damage is sensed and translated into ESCRT recruitment is poorly understood. Here, we identify the two ATG8 E3-like ligases, ATG16L1 and TECPR1, as ion-dependent catalysts for ESCRT recruitment to damaged lysosomal membranes. Leakage from perforated lysosomes induces the proton sensitive V-ATPase-dependent recruitment of ATG16L1-ATG5-ATG12 complexes, or the calcium-sensitive sphingomyelin-dependent recruitment of TECPR1-ATG5-ATG12 complexes. In both cases, the E3-like complex-dependent ATG5-ATG12 conjugate is required for ESCRT recruitment to the damaged membrane, and stabilization of the ESCRT machinery. Collectively, this study establishes the ATG8 E3-like ligases as membrane damage sensors for ESCRT-mediated membrane repair.**

**Keywords** ATG8 E3-like Ligases; ESCRT; Lysosomal Membrane Integrity; Membrane Damage Sensor; CASM
**Subject Categories** Autophagy & Cell Death; Organelles; Post-translational Modifications & Proteolysis

## Introduction

Membranes of the endolysosomal system face frequent damage from pathogenic, chemical or physical stress. As a result, cells have evolved sophisticated strategies to rapidly detect and repair perforated membranes. Central to this response is the endosomal sorting complex required for transport (ESCRT) machinery. This set of multisubunit protein complexes (ESCRT-0, ESCRT-I, ESCRT-II, and ESCRT-III) play an important role in membrane remodeling, and were more recently shown to play a key role in the sealing of damaged endolysosomal membranes (Radulovic et al, 2018; Skowyra et al, 2018). The mechanism behind ESCRT-mediated membrane repair is still not fully understood, but is dependent on ESCRT-III filaments in combination with the ESCRT-III associated protein, ALIX, and ESCRT-I protein, TSG101 (Chen et al, 2019; Radulovic et al, 2018; Skowyra et al, 2018).

A fundamental question pertaining to endolysosomal membrane repair that has yet to be resolved is how the ESCRT machinery senses, and is recruited to, sites of membrane damage. Calcium efflux, in conjunction with the $Ca^{2+}$ binding protein Apoptosis Linked Gene-2 (ALG-2), has been proposed to play a central role in the sensing of damage. Lysosomal membrane damage causes $Ca^{2+}$ to leak out of the lysosome creating a localized increase in cytosolic $Ca^{2+}$ surrounding the damage site. Binding of ALG-2 to $Ca^{2+}$ within this region causes ALG-2 to undergo a conformational change that promotes its interaction with ALIX (Missotten et al, 1999; Suzuki et al, 2008) and TSG101 (Katoh et al, 2005). Thus, ALG-2 is largely considered to be the $Ca^{2+}$-dependent sensor that initiates ESCRT recruitment to sites of damage or osmotic stress (Chen et al, 2024). However, despite ALG-2's inherent membrane binding ability (Shukla et al, 2022), recent reports have shown that ALG-2 membrane binding mutants are still recruited to lysosomes in response to damage (Shukla et al, 2024) suggesting that ALG-2 recruitment may not be the initiating event in ESCRT-mediated repair. In this study, we identify the autophagy E3-like ligase complexes as the bona fide sensors of lysosomal membrane damage.

Macroautophagy (hereafter autophagy) has been linked to the cellular response to membrane damage as a mechanism to sequester and degrade endomembranes that have been damaged beyond the point of repair. Extensive membrane damage will lead to endolysosomal rupture, exposing intraluminal glycans to the cytosol. The binding of glycans by a family of β-galactoside-binding lectins (galectins) serves as a platform to recruit the autophagic machinery required for sequestration of the damaged membrane into a double-membraned autophagosome (Chauhan et al, 2016; Maejima et al, 2013; Paz et al, 2010; Thurston et al, 2012). A key event in autophagosome biogenesis and cargo recognition is the conjugation of autophagy-related (ATG)8 family proteins to phosphatidylethanolamine (PE) or phosphatidylserine (PS) on autophagosomal membranes (a process referred to as membrane ATG8ylation (Kumar et al, 2021)). Lipidation of ATG8 proteins occurs via two ubiquitin-like ATG conjugation systems composed of core ATG genes (ATG3, ATG5, ATG7, ATG10, ATG12 and

[1]SciLifeLab, Department of Chemistry, Umeå University, Umeå SE-90187, Sweden. [2]Umeå Centre for Microbial Research, Umeå University, Umeå SE-90187, Sweden.
✉E-mail: dale.corkery@umu.se; yaowen.wu@umu.se

ATG16L1) (Mizushima, 2020). The ATG5-ATG12-ATG16L1 complex acts as the E3-like ligase, in which ATG16L1 recognizes target membranes and recruits the ATG5-ATG12 conjugate to catalyze the ATG8 lipidation reaction. Recently, ATG8 proteins have been shown to be conjugated to various single-membrane compartments (endolysosomal membranes, phagosomes, Golgi compartments and ER) in response to diverse stimuli. These processes are termed Conjugation of ATG8s to Single Membranes (CASM), and are characterized by the involvement of a subset of components from the autophagy machinery (Durgan and Florey, 2022; Galluzzi and Green, 2019). However, the functions of CASM remain largely unknown.

In the context of endolysosomal damage, ATG16L1 is recruited to damaged membranes via interaction with the Vacuolar type ATPase (V-ATPase), activated in response to damage-induced loss of the proton gradient (Fletcher et al, 2018; Xu et al, 2019) (a process referred to as V-ATPase-ATG16L1-induced ATG8 lipidation (VAIL) (Fischer et al, 2020)). Recently, we and others identified a second E3-like ligase complex which utilizes the autophagosome-lysosome tethering factor, Tectonin beta-propeller repeat containing 1 (TECPR1), in place of ATG16L1 (Boyle et al, 2023; Corkery et al, 2023; Kaur et al, 2023). TECPR1 recognizes damaged membranes via interaction with sphingomyelin, a membrane lipid that translocates from the luminal to cytoplasmic membrane surface in response to damage (Niekamp et al, 2022) (a process that we call sphingomyelin-TECPR1-induced ATG8 lipidation (STIL)). Surprisingly, we found that double knockout of both E3-like ligase complexes compromised membrane repair, suggesting that the role of autophagy proteins in the cellular response to membrane damage may extend beyond autophagic removal.

In this study, we show that E3-like ligase translocation to damaged membranes is a prerequisite for ESCRT recruitment. We demonstrate that this recruitment is dependent on the ATG5-ATG12 conjugate, through both an ATG8ylation-dependent and –independent function.

## Results

### Loss of ATG16L1- and TECPR1-dependent ATG8 E3-like ligase complexes prevents ESCRT recruitment to damaged lysosomes

We and others recently identified a TECPR1-dependent autophagy E3-like ligase complex which functions independently of ATG16L1 to regulate unconventional ATG8 lipidation at damaged lysosomal membranes (Boyle et al, 2023; Corkery et al, 2023; Kaur et al, 2023). Double knockout of both ATG16L1 and TECPR1 was shown to compromise the repair of damaged membranes, suggesting a functionally redundant role for both E3-like ligase complexes in the repair process (Corkery et al, 2023; Corkery and Wu, 2024). To determine the mechanism through which the E3-like ligase complexes contribute to membrane repair, we first assessed ESCRT machinery recruitment to lysosomes damaged by the lysosomal-membrane-damaging agent L-leucyl-L-leucine O-methyl ester (LLOMe), in HeLa (Figs. 1A–C and EV1A,B) and HEK (Fig. EV1C,D) cells deficient for ATG16L1 and/or TECPR1. TECPR1/ATG16L1 double knockout (E3-DKO) cells failed to

recruit the ESCRT III binding protein ALIX to damaged lysosomes, despite abundant membrane damage, indicated by an accumulation of the β-galactoside-binding lectin, Galectin-3 (Gal3). Knockout of either TECPR1 or ATG16L1 alone did not prevent ALIX recruitment, consistent with a functional redundancy between the two complexes. Immunostaining for ESCRT-III complex members IST1 and CHMP2A confirmed that the ESCRT-III-dependent membrane repair complex is absent from damaged membranes in E3-DKO cells (Fig. 1D). Furthermore, live-cell imaging of HeLa cells co-transfected with IST1-EGFP and LAMP1-mCherry confirmed impaired IST1 recruitment to lysosomes following LLOMe treatment in the absence of the ATG8 E3-like ligases (Fig. 1E).

To determine if the impaired ESCRT recruitment translated into increased susceptibility to damage, HeLa WT and E3-DKO cells were treated with low-dose LLOMe (250 μM) and immunostained for ALIX and Gal3 at 0, 5, 10, 15, and 20 min post treatment (Appendix Fig. S1). E3-DKO cells showed accelerated Gal3 recruitment, as compared to WT, suggesting an increased susceptibility to rupture. Importantly, the recently identified phosphoinositide-initiated membrane tethering and lipid transport (PITT) pathway for lysosomal repair (Tan and Finkel, 2022) appears unaffected by the loss of the ATG8 E3-like ligases, as damage-induced lysosomal PI4P accumulation was evident in E3-DKO cells treated with LLOMe (Appendix Fig. S2).

To further characterize the mechanism behind E3-like ligase-dependent ESCRT recruitment, E3-DKO addback cell lines were generated which stably express wild-type (WT) TECPR1, or one of two TECPR1 mutants. TECPR1$^{\Delta1-377}$ is lacking the N-terminal dysferlin domain required for lysosomal translocation in response to membrane damage (Corkery et al, 2023). TECPR1$^{\Delta AIR}$ is lacking the ATG5 interaction region (AIR) (Chen et al, 2012), preventing co-recruitment of ATG5 to the damaged membrane (Appendix Fig. S3). Damage-induced lysosomal ALIX recruitment was restored with the addback of wild-type TECPR1, but not with either TECPR1 mutant (Fig. 1F–I), suggesting that TECPR1-dependent recruitment of ATG5 to the damaged membrane is required for subsequent ESCRT recruitment. A recent study reported that loss of ATG5 impaired ESCRT recruitment to damaged lysosomes due to an increase in the ATG12-ATG3 sidestep conjugate with an affinity for ALIX (Wang et al, 2023). The authors propose that, in the absence of its preferred conjugation partner (ATG5), ATG12 is free to form an alternative conjugate with ATG3. While we do observe reduced ATG5-ATG12 conjugate expression in E3-DKO cells (Fig. EV1D), the addback of TECPR1$^{WT}$ or the lysosome-binding-deficient TECPR1$^{\Delta1-377}$ was sufficient to restore conjugate expression (Fig. 1G). TECPR1$^{\Delta AIR}$ did not restore conjugate expression suggesting conjugate stability/regulation is tied to E3-like ligase complex assembly (Fig. 1G). Importantly, despite its ability to restore ATG5-ATG12 conjugate expression, TECPR1$^{\Delta1-377}$ did not restore ALIX recruitment, indicating that ESCRT recruitment is dependent on ATG5 translocation to the damaged membrane.

### ESCRT recruitment to damaged lysosomes can occur independent of ATG8 lipidation

Within the trimeric E3-like ligase complexes, the ATG5-ATG12 conjugate possesses the E3-like ligase activity required for the ATG8-PE or -PS conjugation reaction (Hanada et al, 2007). Our

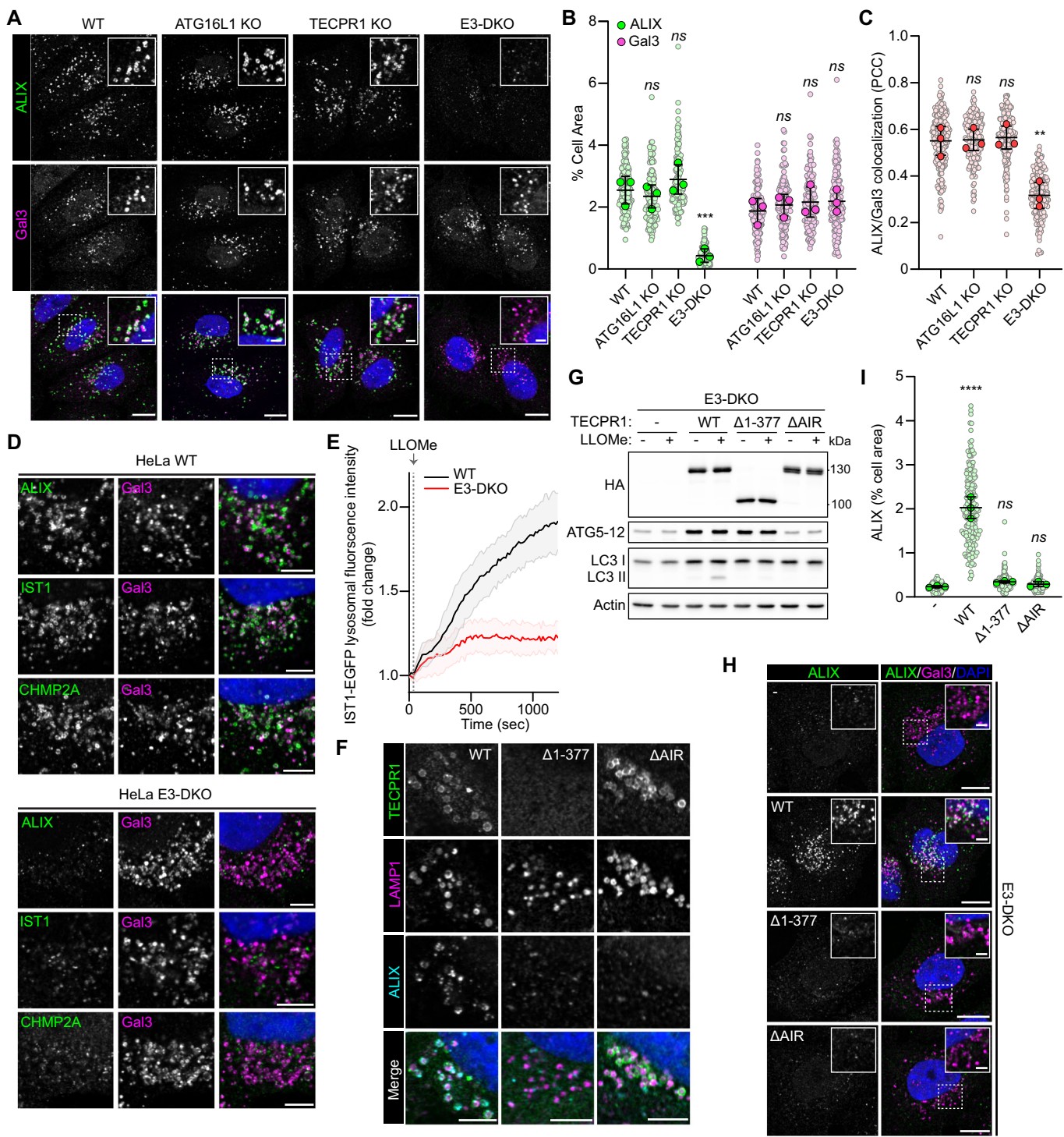

observation that ATG5 recruitment to damaged membranes is an essential prerequisite to ESCRT recruitment therefore suggests that ATG8ylation of the damaged membrane may be a contributing factor in this recruitment. To confirm this hypothesis, we evaluated the competency of ESCRT recruitment in cell lines deficient for either ATG5 or the E1-like enzyme required for the ATG5-ATG12 conjugation reaction, ATG7 (Fig. EV2A,B). Both cell lines are deficient for the ATG5-ATG12 conjugate (Fig. EV2B) and both

failed to recruit ALIX to lysosomes damaged by LLOMe (Fig. 2A,B). To determine if the E3-like ligase activity of the ATG5-ATG12 conjugate was required, we employed hexa-KO HeLa cells lacking the six human ATG8 paralogues (LC3A, LC3B, LC3C, GABARAP, GABARAPL1 and GABARAPL2) (ATG8 KO) (Nguyen et al, 2016), or HeLa cells deficient for the cysteine proteases required for ATG8 processing, ATG4 A/B/C/D (Nguyen et al, 2021). Both cell lines continue to express the ATG5-ATG12 conjugate (Fig. EV2B) but

**Figure 1. ESCRT recruitment to damaged lysosomes is impaired in cell lines deficient for the autophagy E3-like complexes.**

(A) Confocal images of HeLa WT, ATG16L1 KO, TECPR1 KO and E3-DKO cells treated with 1 mM LLOMe for 30 min. Nuclei were stained with DAPI. Scale bars = 10 μm. (B, C) Quantification of ALIX and Gal3 area (B) and colocalization (C) from (A). Small points represent individual cells from three independent experiments. Large points represent the means of individual experiments ($n \geq 50$ cells per experiment). Bars represent the mean ± SD from the three experiments. Significance was determined from biological replicates using a one-way ANOVA with Tukey's multiple comparisons tests. (B) ns (not significant) represents $P > 0.05$, ***$P = 0.0001$. (C) ns (WT vs 16KO) $P = 0.9997$, ns (WT vs TECKO) $P = 0.9873$, **$P = 0.0030$. (D) Confocal images of HeLa WT and E3-DKO cells treated with 1 mM LLOMe for 30 min. Scale bars = 5 μm. (E) Quantification of IST1 lysosomal recruitment in HeLa WT and E3-DKO cells co-transfected with IST1-EGFP and LAMP1-mCherry and treated with 1 mM LLOMe. Images were acquired every 15 s. Data are presented as mean ± SD from five independent experiments ($n > 25$ cells). (F) Confocal images of HeLa TECPR1/ATG16L1 DKO cells transfected with the indicated TECPR1 mutant and treated with 1 mM LLOMe for 30 min. Scale bars = 5 μm. (G) Western blot analysis of HeLa TECPR1/ATG16L1 DKO cells stably expressing the indicated TECPR1 mutant and treated +/− LLOMe. (H) Confocal images of cell lines from (E) treated with 1 mM LLOMe for 30 min. Scale bars = 10 μm for whole cell images and 2 μm for insets. (I) Quantification of ALIX area from (H). Small points represent individual cells from three independent experiments. Large points represent the means of individual experiments ($n = 60$ cells per experiment). Bars represent the mean ± SD from the three experiments. Significance was determined from biological replicates using a one-way ANOVA with Tukey's multiple comparisons tests. ns (− vs Δ1-377) $P = 0.7197$, ns (− vs ΔAIR) $P = 0.9439$, ****$P < 0.0001$. Source data are available online for this figure.

are unable to generate lipidated ATG8. Surprisingly, we observed significant lysosomal ALIX recruitment in ATG8 KO and ATG4 KO cell lines following treatment with LLOMe (Fig. 2A,B). These data suggest that, despite the requirement for ATG5-ATG12 conjugation, ATG5's role in ESCRT recruitment may not be solely linked to membrane ATG8ylation. To explore further, we stably introduced ATG5 or the ATG5$^{K130R}$ mutant which cannot undergo ATG12 conjugation into ATG5 KO cells (Fig. 2C). Damage-induced lysosomal ALIX recruitment was restored with the addback of WT ATG5, but not with the conjugation-deficient mutant (Fig. 2D), confirming a specific requirement for the ATG5-ATG12 conjugate at the damaged membrane.

## Membrane ATG8ylation is required for ESCRT-mediated repair

A recent report provided evidence suggesting that non-canonical lipidation of GABARAPs is essential for ESCRT recruitment to damaged lysosomes. The authors observed reduced ESCRT recruitment in GABARAP TKO cells, which they attributed to a direct interaction between GABARAPL2 and ALIX (Ogura et al, 2023). In contrast to ATG5-ATG12 deficient cell lines, in which ALIX recruitment was completely abolished, we observed significant ALIX translocation to damaged lysosomes in ATG8 KO cells (Fig. 2A,B). To determine how this translocation compared to wild-type cells, we performed immunostaining for ESCRT-III proteins IST1 and CHMP2A in HeLa WT, ATG5 KO and ATG8 KO cells in which lysosomes were damaged by LLOMe treatment (Fig. 3A,B). Similar to ALIX, we observed increased IST1/CHMP2A puncta formation in ATG8 KO cells, as compared to ATG5 KO cells. However, in agreement with the above mentioned report, ESCRT translocation in ATG8 KO cells was significantly less than in wild-type cells, and could be rescued with the addback of GABARAPL2 (Appendix Fig. S4). This suggests that there could be both a conjugation-dependent and conjugation-independent role for ATG5-ATG12 in regulating ESCRT recruitment to damaged membranes.

To assess the impact of membrane ATG8ylation on ESCRT recruitment, we performed high-resolution (Appendix Fig. S5) or super-resolution structured illumination (Fig. 3C) imaging of the ESCRT machinery on damaged lysosomes in wild-type and ATG8 KO cells. In wild-type cells, the ESCRT machinery appears evenly distributed on lysosomes damaged by LLOMe treatment. In contrast,

cell lines deficient for ATG8ylation display a much more fragmented or incomplete ESCRT distribution, which likely explains the differences in ESCRT recruitment observed between the two cell lines (Fig. 3A,B). To better observe the architecture of the ESCRT machinery on damaged vesicles, we transfected cells with constitutively active Rab5 (Rab5$^{Q79L}$), shown to promote endosomal fusion resulting in the formation of oversized early/late endosome-like vesicles (Bucci et al, 1992; Stenmark et al, 1994). LysoTrackerRED staining of HeLa cells stably expressing EGFP-Rab5$^{Q79L}$ confirmed the presence of large acidified vesicles (Fig. EV3A) which we hypothesized, due to the low pH, could be susceptible to damage by lysosomotropic compounds like LLOMe. To confirm, cells were transfected with mCherry-TECPR1 and treated with 0.5 mM LLOMe for 10 min (Fig. EV3B,C). Similar to what we have observed with lysosomes (Corkery et al, 2023), TECPR1 was rapidly recruited to the oversized vesicles within minutes of LLOMe addition, confirming LLOMe-induced membrane damage. Due to the enlarged size of the vesicles, we were able to clearly differentiate ATG8 E3-like ligase, ATG8, and ESCRT recruitment to the surface of the damaged membrane, from Gal3 recruitment to intraluminal glycans (Fig. EV3D)—thus providing a platform to assess the architecture of membrane repair proteins using traditional confocal microscopy. To determine the impact of membrane ATG8ylation on ESCRT machinery architecture at damaged membranes, Rab5$^{Q79L}$ was stably expressed in HeLa WT, ATG8 KO and ATG5 KO cells, and membranes of the oversized vesicles were damaged with LLOMe. Immunostaining for IST1 revealed a uniform distribution of the ESCRT machinery on damaged membranes in wild-type cells (Fig. 3D), which was unaffected by the single knockout of either TECPR1 or ATG16L1 (Fig. EV3E,F). In contrast, damaged vesicles in ATG8 KO cells had reduced ESCRT recruitment with a more fragmented architecture, while vesicles in ATG5 KO cells failed to recruit the ESCRT machinery entirely (Fig. 3D,E).

To determine how the altered ESCRT architecture at damaged membranes affected repair, we first assessed susceptibility to lysosomal rupture in HeLa WT, E3-DKO, ATG5 KO and ATG8 KO cells. Nanoscale damage to lysosomal membranes causes ion leakage which is believed to be the driving force for ESCRT recruitment. More extensive damage leads to the formation of larger pores or rupturing of the limiting membrane, causing permeability for proteins. Thus, the recruitment of Gal3 to luminal β-galactosides can be used as a marker of extensive lysosomal damage (Skowyra et al, 2018). Therefore, to assess sensitivity to

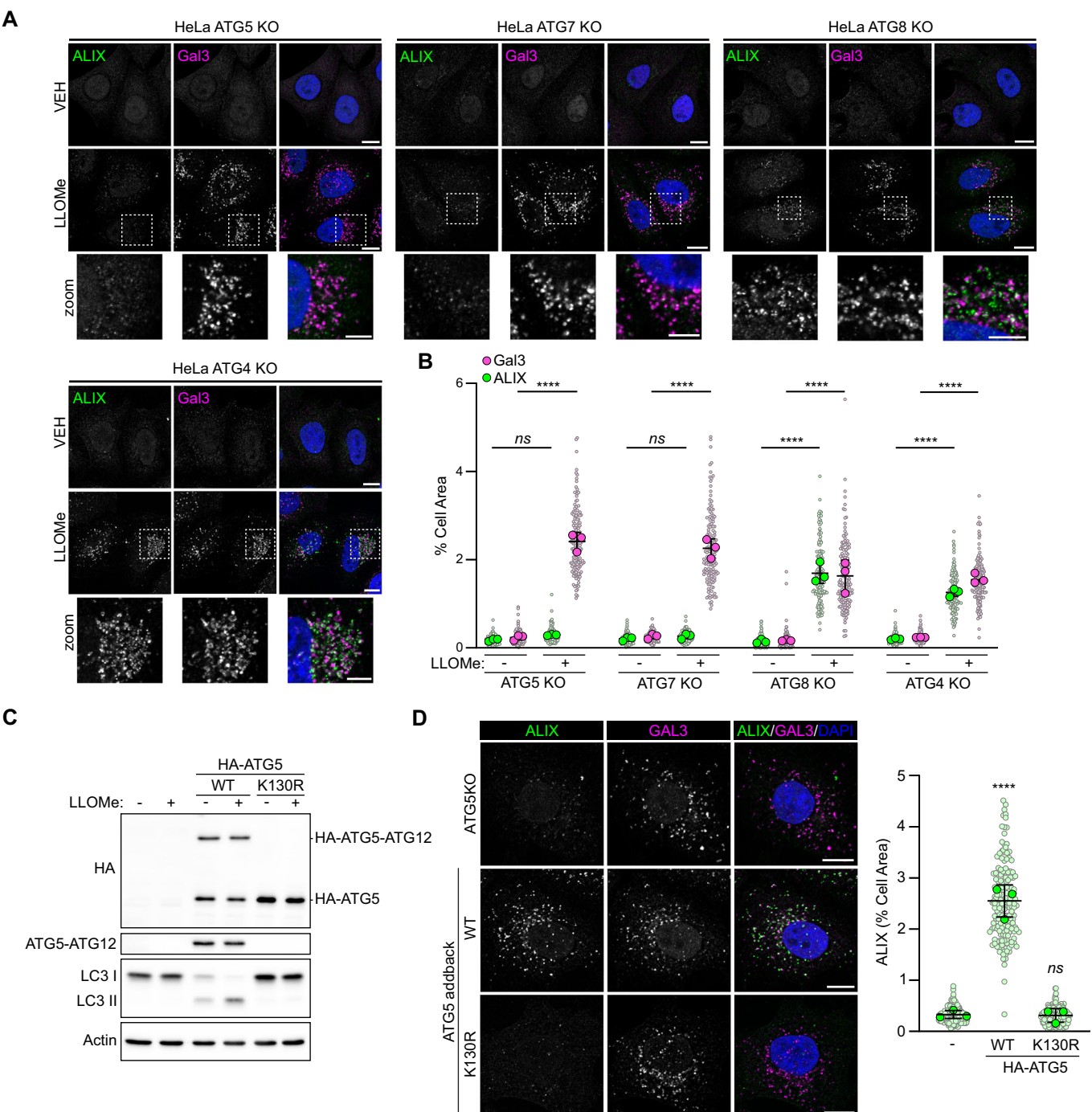

**Figure 2. ESCRT recruitment to damaged lysosomes can occur without ATG8 lipidation.**

(A) Confocal images of ATG KO cell lines treated with or without 1 mM LLOMe for 30 min and immunostained for ALIX and Gal3. Scale bars = 10 μm for whole image and 5 μm for insets. (B) Quantification of ALIX and Gal3 area from (A). Small points represent individual cells from three independent experiments. Large points represent the means of individual experiments ($n = 50$ cells per experiment). Bars represent the mean ± SD from the three experiments. Significance was determined from biological replicates using a one-way ANOVA with Tukey's multiple comparisons tests. ns (5KO) $P > 0.9999$, ns (7KO) $P > 0.9999$, ****$P = < 0.0001$. (C) Western blot analysis of ATG5 KO cells stably expressing WT HA-ATG5 or HA-ATG5[K130R]. (D) Confocal images of cell lines from (C) treated with 1 mM LLOMe for 30 min and immunostained for ALIX and Gal3. Scale bars = 10 μm. Quantification of ALIX area is shown to the right. Small points represent individual cells from three independent experiments. Large points represent the means of individual experiments ($n = 60$ cells per experiment). Bars represent the mean ± SD from the three experiments. Significance was determined from biological replicates using a one-way ANOVA with Tukey's multiple comparisons tests. ns not significant ($P = 0.9930$), ****$P < 0.0001$. Comparisons against WT are shown. Source data are available online for this figure.

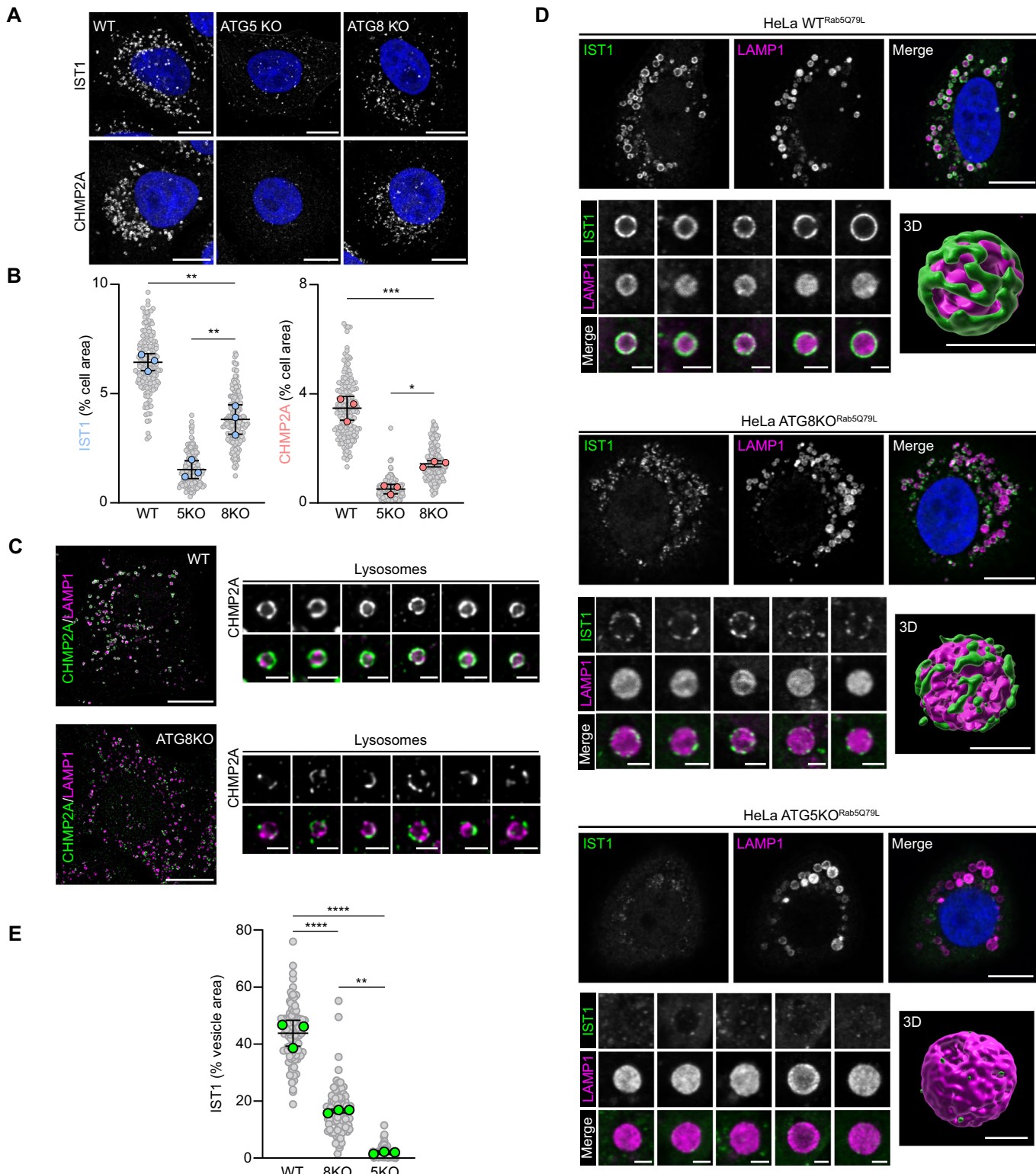

lysosomal rupture, cells were treated with 250 µM LLOMe for 5 min, a dosing regimen shown to induce nanoscale damage (ESCRT recruitment) without rupturing the lysosome (Gal3 recruitment) (Appendix Fig. S1). LLOMe was washed off and the cells allowed to recover for 20 min, followed by immunostaining for

ALIX and Gal3 (Fig. 4A). Double knockout of the E3-like ligases, or knockout of ATG5, resulted in a significant increase in Gal3 staining after LLOMe washout, suggesting that impaired ESCRT recruitment allowed for the nanoscale damage to progress to full lysosomal rupture (Fig. 4B). In contrast, limited Gal3 staining

**Figure 3.  ATG8 lipidation is required for complete ESCRT recruitment to damaged lysosomes.**

(A) Confocal images of HeLa WT, ATG5 KO and ATG8 KO cells treated with 1 mM LLOMe for 20 min. Scale bars = 10 μm. (B) Quantification of IST1 and CHMP2A cell area from (A). Small points represent individual cells from three independent experiments. Large points represent the means of individual experiments (n > 50 cells per experiment). Bars represent the mean ± SD from the three experiments. Significance was determined from biological replicates using a one-way ANOVA with Tukey's multiple comparisons tests. *P = 0.0156, **(WT vs 8KO) P = 0.0017, ** (8KO vs 5KO) P = 0.0034, ***P = 0.0003. (C) Super-resolution structured illumination images of HeLa WT and ATG8 KO cells treated with 1 mM LLOMe for 20 min. Scale bars = 10 μm for whole cell images and 1 μm for individual lysosome images. (D) Representative confocal images of HeLa WT, ATG5 KO and ATG8 KO cells transfected with Rab5$^{Q79L}$ and treated with 0.5 mM LLOMe for 20 min. Scale bars = 10 μm for whole cell images and 2 μm for individual vesicle images. (E) Quantification of IST1 vesicle area from (D). Grey points represent individual vesicles from three independent experiments (n ≥ 30 lysosomes per experiment). Bars represent the mean ± SD from the three experiments. Significance was determined from biological replicates using a one-way ANOVA with Tukey's multiple comparisons tests. **P = 0.0013, ****P = <0.0001. Source data are available online for this figure.

was observed in ATG8 KO cells, suggesting the reduced ESCRT recruitment observed in those cells was sufficient to provide protection against lysosomal rupture (Fig. 4B).

To determine if this increased resistance to lysosomal rupture was due to active repair of damaged lysosomes, we performed a lysosome re-acidification assay in the same panel of cell lines. Lysosomes were loaded with LysoTrackerRED and pulsed with 250 μM LLOMe for 10 min to induce damage. LLOMe was washed away, and cells were allowed to recover in LysoTrackerRED containing media for 45 or 90 min. Restoration of LysoTrackerRED staining was used as an indicator of successful lysosome re-acidification (Fig. 4C). As previously reported, double knock out of the autophagy E3-like ligases significantly impaired lysosome re-acidification after damage (Fig. 4D) (Corkery et al, 2023). A similar impairment was observed in ATG5 KO cells, consistent with our observation that E3-like ligase-dependent ATG5-ATG12 recruitment is a prerequisite for ESCRT assembly. Interestingly, lysosomal re-acidification in ATG8 KO cells was as inefficient as in ATG5 KO or E3-DKO cells, suggesting that, despite sufficient ESCRT recruitment to provide resistance against rupture, the repair of damaged lysosomes remains impaired in the absence of membrane ATG8ylation.

## ALG-2 recruitment to damaged membranes is dependent on the ATG8 E3-like ligases

ALG-2 has been proposed to sense Ca$^{2+}$ released into the cytosol from perforated lysosomes and recruit the ESCRT machinery to sites of damage via direct interaction with ALIX and TSG101 (Chen et al, 2024; Shukla et al, 2024). To determine if E3-like ligase recruitment to damaged membranes influences ALG-2, HeLa WT, ATG16L1 KO, TECPR1 KO, E3-DKO, ATG5 KO and ATG8 KO cells were treated with LLOMe and immunostained for ALG-2 (Fig. 5A,B). Consistent with previous reports, we observed ALG-2 accumulation at damaged lysosomes in wild-type cells (Skowyra et al, 2018). Knockout of ATG16L1 or TECPR1 alone had no impact on ALG-2 recruitment, while knockout of both E3-like ligases (or ATG5) blocked ALG-2 translocation to damaged membranes (Fig. 5A,B; Appendix Fig. S6). This recruitment was shown to be dependent on the ATG5-ATG12 conjugate as addback of wild-type ATG5 to ATG5 KO cells restored ALG-2 transloca-tion, while addback of the conjugation-deficient ATG5$^{K130R}$ did not (Fig. 5C). LLOMe-induced ALG-2 puncta were enriched for ATG5 (Fig. 5D) further suggesting that E3-like ligase translocation to damaged membranes acts to recruit ALG-2. In the absence of ATG8s, ALG-2 recruitment to damaged lysosomes was impaired (Fig. 5A,B; Appendix Fig. S6), suggesting that membrane ATG8ylation contributes to ALG-2 recruitment.

## ALG-2 is dispensable for ESCRT recruitment to damaged membranes

Due to its ability to bind calcium, ALG-2 is thought to be responsible for sensing lysosomal membrane permeabilization and translating this signal into the recruitment of the repair machinery (Chen et al, 2024; Shukla et al, 2024; Skowyra et al, 2018). However, recent studies have suggested that this dependence on ALG-2 may vary depending on the type of stress applied to membranes (Yim et al, 2022). To determine if ALG-2 is required for ESCRT recruitment in response to the LLOMe-induced membrane damage used in this study, HeLa WT and ALG-2 KO cells were treated with 0.5 mM LLOMe and ESCRT translocation to damaged membranes assessed by immunofluorescence (Fig. 6A–D). Loss of ALG-2 did not prevent IST1 recruitment to lysosomal membranes damaged with LLOMe suggesting that, in this context, ALG-2 does not function as the sensor of membrane damage.

To determine if ALG-2 instead plays a functional role in the repair process, we assessed susceptibility to lysosomal rupture in HeLa WT and ALG-2 KO cells using the above-described Gal3-based rupture assay (Fig. 4A). ALG-2 KO cells displayed a significant increase in Gal3 staining after LLOMe washout, as compared to WT, confirming an increased susceptibility to lysosome rupture in the absence of ALG-2 (Fig. 6E,F). Therefore, in the case of LLOMe-induced membrane damage, although ALG-2 is required for membrane repair, it is the ATG8 E3-like ligases, not ALG-2, which function as the sensors of lysosomal membrane permeabilization.

The ability to recruit the ESCRT machinery to LLOMe-damaged lysosomes in the absence of ALG-2 raises questions regarding the function of calcium in this process. Calcium chelation has been shown to prevent ESCRT recruitment in response to LLOMe treatment (Skowyra et al, 2018), which is often attributed to the ESCRT-binding ability of calcium-activated ALG-2 (Shukla et al, 2022). To explore the requirement for calcium in our system, HeLa WT, ATG5 KO, E3-DKO and ATG8 KO cells were pre-treated with or without the membrane permeable calcium chelator BAPTA-AM prior to the induction of lysosomal membrane damage with LLOMe. In WT cells, a 45-minute pre-treatment with 25 μM BAPTA-AM did not significantly block CHMP2A recruitment to lysosomes damaged with 1 mM LLOMe for 30 min (Fig. EV4A,B). To confirm that the calcium chelation was successful, we performed immunostaining for ALG-2 and ALIX in cells treated with the same chelation protocol (Fig. EV5). Pre-treatment with BAPTA-AM prevented ALG-2 recruitment to damaged lysosomes, confirming successful calcium chelation, but did not inhibit the recruitment of ALIX, providing further support for the dispensability of ALG-2 in LLOMe-induced ESCRT recruitment. Interestingly, the residual ESCRT recruitment we observe in ATG8 KO cells is abolished by

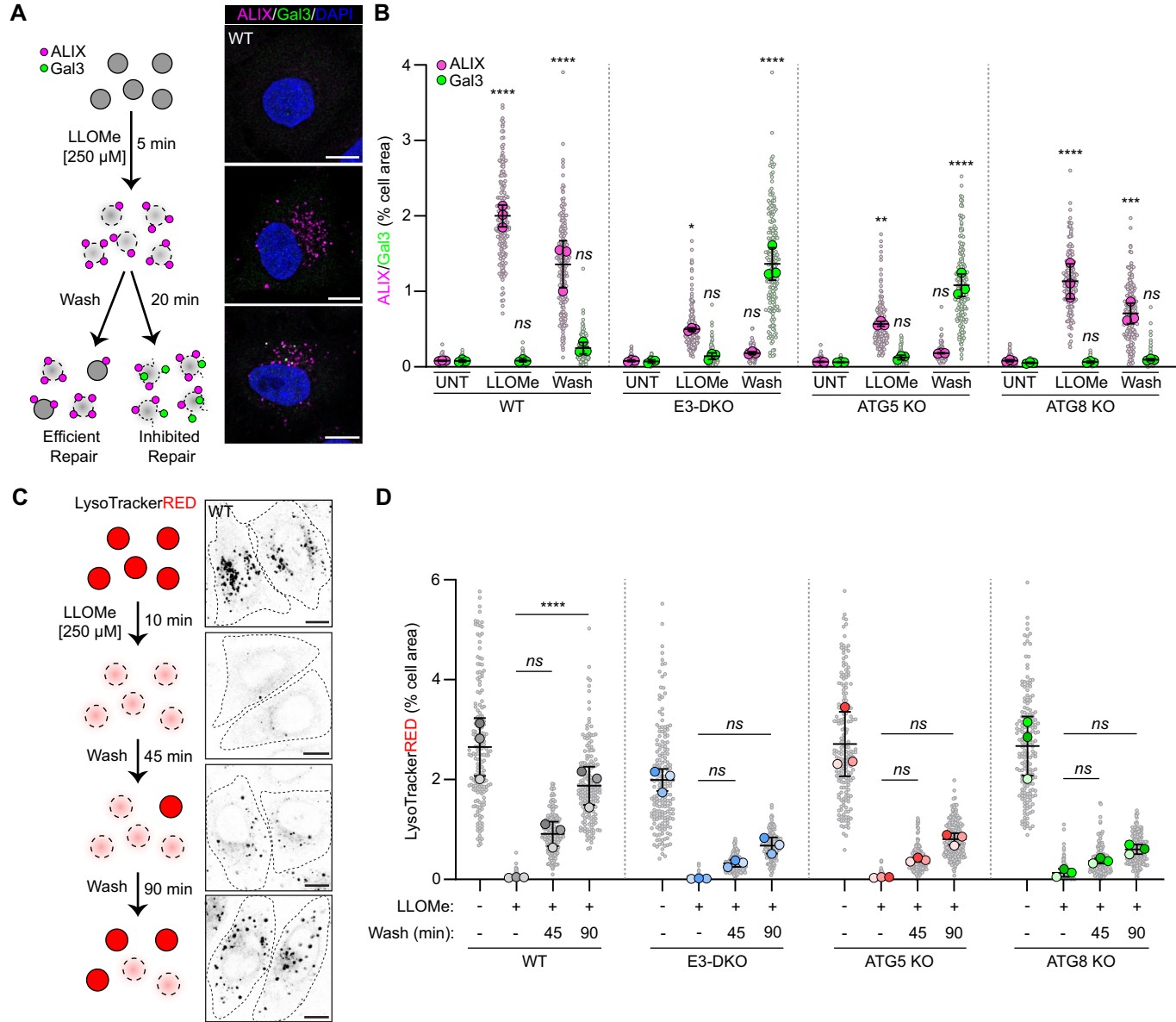

**Figure 4. Membrane ATG8ylation is required for ESCRT-mediated repair.**

(A) Schematic outline of lysosome rupture assay. Scale bars = 10 μm. (B) Quantification of ALIX and Gal3 area from (A). Small points represent individual cells from three independent experiments. Large points represent the means of individual experiments (n = 60 cells per experiment). Bars represent the mean ± SD from the three experiments. Significance was determined from biological replicates using a one-way ANOVA with Tukey's multiple comparisons tests. Comparisons are shown against the cell line-matched untreated sample. ns (not significant) represents P > 0.05 (WT-LLOMe-Gal3, P > 0.9999; WT-Wash-Gal3, P = 0.3449; E3DKO-LLOMe-Gal3, P = 0.9935; E3DKO-Wash-ALIX, P = 0.9962; 5KO-LLOMe-Gal3, P = 0.9976; 5KO-Wash-ALIX, P = 0.9919; 8KO-LLOMe-Gal3, P > 0.9999; 8KO-Wash-Gal3, P > 0.9999), *P = 0.0191, **P = 0.0028, ***P = 0.0001, ****P < 0.0001. (C) Schematic outline of lysosome repair/re-acidification assay. Scale bars = 10 μm. (D) Quantification of LysoTracker Red puncta from (C). Grey points represent individual cells from three independent experiments. Colored points represent the means of individual experiments (n = 60 cells per experiment). Bars represent the mean ± SD from the three experiments. Significance was determined from biological replicates using a one-way ANOVA with Tukey's multiple comparisons tests. ns (not significant) represents P > 0.05 (WT 45 min, P = 0.0654; E3DKO 45 min, P = 0.9955; E3DKO 90 min, P = 0.3624; 5KO 45 min, P = 0.9825; 5KO 90 min, P = 0.1703; 8KO 45 min, P = 0.9996; 8KO 90 min, P = 0.8363), ****P < 0.0001. Source data are available online for this figure.

calcium chelation (Fig. EV4A,B), suggesting that membrane ATG8ylation-independent ESCRT recruitment is calcium-dependent.

ATG8 E3-like ligase recruitment to damaged lysosomes is initiated either by collapse of the proton gradient (ATG16L1-V-ATPase), or by damage-induced sphingomyelin (SM) scrambling at the lysosomal membrane (TECPR1-SM). The scrambling of SM was shown to be calcium-dependent (Niekamp et al, 2022), suggesting that the recruitment of the TECPR1-ATG5-ATG12 E3-like complex to damaged membranes would be sensitive to calcium chelation. To confirm, HeLa WT, ATG16L1 KO and TECPR1 KO cells were pre-treated with or without BAPTA-AM, lysosome damage induced with

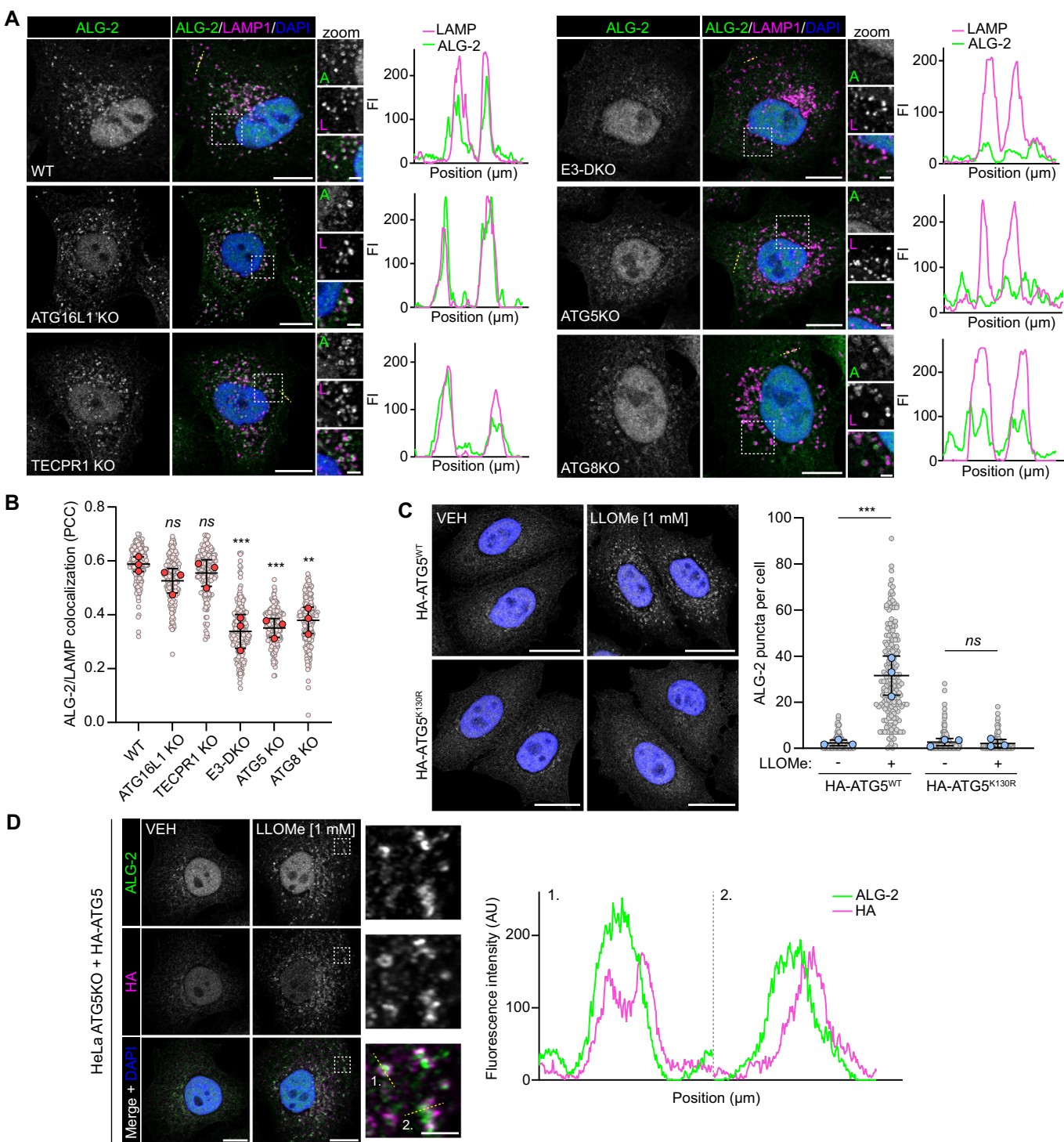

LLOMe and immunostaining performed for CHMP2A (Fig. EV4C,D). Of the three cell lines, only ATG16L1 KO cells, where the SM-TECPR1 axis is functional, showed reduced ESCRT recruitment in response to calcium chelation. This data suggests that the calcium-dependence of ESCRT recruitment to damaged membranes could be partially attributed to SM-dependent recruitment of the TECPR1 E3-like ligase complex. It also further confirms that the two E3-like complexes play a functionally redundant role at the damaged membrane, as inhibition

of the TECPR1 E3-like complex with calcium chelation in WT cells appears to be compensated for by the presence of ATG16L1.

## Discussion

In this study, we show that the ATG8 E3-like ligases function as damage sensors, playing an essential role in ESCRT machinery

**Figure 5. ALG-2 recruitment to damaged membranes is dependent on the E3 ligases.**

(A) Confocal images of HeLa WT, ATG16L1 KO, TECPR1 KO, E3-DKO, ATG5 KO and ATG8 KO cells treated with 1 mM LLOMe for 15 min. Scale bars = 10 μm for whole cell images and 2 μm for insets. Corresponding fluorescence intensity profiles are shown to the right (location marked by a yellow line on the cell image). (B) Colocalization analysis of data presented in (A). Small points represent individual cells from three independent experiments. Large blue points represent the means of individual experiments (n > 50 cells per experiment). Bars represent the mean ± SD from the three experiments. Significance was determined from biological replicates using a one-way ANOVA with Tukey's multiple comparisons tests. ns (not significant) represents $P > 0.05$, $**P = 0.0014$, $***P = 0.0004$. (C) Confocal images of ATG5 KO cells stably expressing WT HA-ATG5 or HA-ATG5$^{K130R}$ treated with or without LLOMe for 15 min. Scale bars = 20 μm. Quantification of cytosolic ALG-2 puncta is shown to the right. Small grey points represent individual cells from three independent experiments. Large blue points represent the means of individual experiments (n > 57 cells per experiment). Bars represent the mean ± SD from the three experiments. Significance was determined from biological replicates using a one-way ANOVA with Tukey's multiple comparisons tests. ns (not significant) represents $P > 0.05$, $***P = 0.0002$. (D) Confocal images of ATG5 KO cells stably expressing WT HA-ATG5, treated with or without LLOMe for 15 min. Scale bars = 10 μm for whole cell images and 2 μm for insets. Corresponding fluorescence intensity profiles are shown to the right (location marked by a yellow line on the inset image). Source data are available online for this figure.

recruitment to lysosomal membranes damaged with LLOMe. Small perforations in the lysosomal membrane result in the leakage of both protons and $Ca^{2+}$ into the cytosol. Collapse of the proton gradient induces assembly of the V-ATPase proton pump on lysosomal membranes for the purpose of restoring the gradient (Mulligan et al, 2024). V-ATPase subsequently recruits the ATG16L1-ATG5-ATG12 E3-like ligase complex (Xu et al, 2019) to the damaged membrane. $Ca^{2+}$ leakage induces the scrambling of sphingomyelin (SM) from the luminal to cytoplasmic membrane surface of the lysosomal membrane (Niekamp et al, 2022). SM subsequently recruits the TECPR1-ATG5-ATG12 E3-like ligase to the damaged membrane (Boyle et al, 2023; Corkery et al, 2023; Kaur et al, 2023). Here we show that ATG8 E3-like ligase translocation is an essential prerequisite to ESCRT recruitment, thereby placing the ATG8 E3-like ligases in the role of damage sensors for the ESCRT-mediated membrane repair pathway. We demonstrate that ESCRT recruitment is dependent on the ATG5-ATG12 conjugate, which plays both an ATG8 lipidation-dependent and ATG8 lipidation-independent role in regulating ESCRT recruitment to the damaged membrane.

Calcium leakage from damaged lysosomes is proposed to be the driving force behind ESCRT recruitment (Chen et al, 2024; Shukla et al, 2022; Skowyra et al, 2018) due to direct interaction between $Ca^{2+}$-activated ALG-2 and several components of the ESCRT complex (ALIX, TSG101, VPS37B/C, IST1) (Katoh et al, 2005; Missotten et al, 1999; Okumura et al, 2013a; Okumura et al, 2013b). Contrary to what has been reported for lysosomal damage induced by Glycyl-L-phenylalanine 2-naphthylamide (GPN) (Chen et al, 2024), we found that ALG-2 is dispensable for ESCRT recruitment to lysosomes damaged by LLOMe. This would suggest that ESCRT recruitment is differentially regulated based on the type of lysosomal stress applied. However, studies utilizing GPN to study ALG-2 at the lysosome should be interpreted with caution as GPN has been shown to also induce $Ca^{2+}$ release from the ER (Atakpa et al, 2019), which promotes additional ALG-2 recruitment to ER exit sites (Shibata et al, 2007; Shukla et al, 2024). The dispensability of ALG-2 during LLOMe-induced damage raises questions regarding the role of calcium in ESCRT recruitment, as several studies have shown that calcium chelation can prevent ESCRT recruitment to lysosomes damaged with LLOMe (Herbst et al, 2020; Jia et al, 2020; Skowyra et al, 2018). Our data suggests that, in the context of LLOMe-induced damage, calcium plays an additional role in the SM-dependent recruitment of the TECPR1-containing E3-like ligase complex. We hypothesize that different tissues/cell lines may favor one E3-like complex over the other, and that this difference could account for the variability observed in the inhibition ESCRT recruitment by calcium chelation.

Recent studies have offered conflicting hypotheses regarding the role of autophagy proteins in ESCRT-mediated membrane repair. Loss of ATG5 has been reported to block ESCRT recruitment to damaged lysosomes due to an increase in the ATG12-ATG3 sidestep conjugate with an affinity for ALIX (Wang et al, 2023). These conclusions were drawn, in part, due to the observation that ATG5 KO, but not ATG16L1 KO, impaired ALIX recruitment to lysosomes damaged with LLOMe. Our recent identification of a second E3-like ligase complex (TECPR1-ATG5-ATG12) which plays a functionally redundant role in lysosome repair (Corkery et al, 2023) has allowed us to expand upon these observations by demonstrating that ATG16L1- and/or TECPR1-mediated ATG5-ATG12 recruitment to the damaged membrane is an essential step in the ESCRT repair pathway. In a second report, non-canonically lipidated ATG8s were shown to support ESCRT recruitment *via* a direct interaction between GABARAPL2 and ALIX (Ogura et al, 2023). While we report significant damage-induced lysosomal enrichment of ALIX in the absence of ATG8s (or ATG4), a comparison against wild-type cells revealed that the localization of ESCRT machinery on damaged lysosomes is distorted in the absence of membrane ATG8ylation. This supports a hypothesis whereby ATG5-ATG12 recruits the ESCRT machinery to the damaged membrane. After recruitment, ESCRT localization and/or function is stabilized *via* multiple interactions with ATG8s conjugated to the damaged membrane. This study demonstrates a fundamental function of CASM involved in ESCRT-mediated membrane repair during lysosomal damage.

## Methods

### Reagents and tools table

| Reagent/resource | Reference or source | Identifier or catalog number |
|---|---|---|
| **Experimental models** | | |
| HeLa WT | Nakamura et al, 2020 | |
| HeLa ATG16L1 KO | Nakamura et al, 2020 | |
| HeLa TECPR1 KO | This study | |
| HeLa E3 DKO | This study | |
| HeLa ATG5 KO | Nakamura et al, 2020 | |
| HeLa ATG7 KO | Nakamura et al, 2020 | |

| Reagent/resource | Reference or source | Identifier or catalog number |
|---|---|---|
| HeLa WT | Nguyen et al, 2016 | |
| HeLa ATG8 KO | Nguyen et al, 2016 | |
| HeLa ATG4 KO | Nguyen et al, 2021 | |
| HeLa WT | Takahara et al, 2017 | |
| HeLa ALG2 KO | Takahara et al, 2017 | |
| HEK293 WT | Lystad et al, 2019 | ATCC CRL-1573 |
| HEK293 ATG16L1 KO | Lystad et al, 2019 | |
| HEK293 TECPR1 KO | Corkery et al, 2023 | |
| HEK293 E3 DKO | Corkery et al, 2023 | |
| **Recombinant DNA** | | |
| EGFP-TECPR1 | Wetzel et al, 2020 | |
| EGFP-TECPR1$^{\Delta 1-377}$ | Corkery et al, 2023 | |
| EGFP-TECPR1$^{\Delta AIR}$ | This study | |
| HA-TECPR1 | This study | |
| HA-TECPR1$^{\Delta 1-377}$ | This study | |
| HA-TECPR1$^{\Delta AIR}$ | This study | |
| LAMP1-mCherry | Corkery et al, 2023 | |
| IST1-EGFP | This study | |
| GFP-P4M-SidM | Addgene | Cat# 51469 |
| pGABARAPL2 | This study | |
| EGFP-Rab5 | This study | |
| EGFP-Rab5$^{Q79L}$ | This study | |
| mCherry-ATG5 | Corkery et al, 2023 | |
| HA-ATG5 | This study | |
| HA-ATG5$^{K130R}$ | This study | |
| pSpCas9(BB)-2A-Puro(PX459) | Addgene | Cat# 62988 |
| **Antibodies** | | |
| Mouse anti-ALIX | BioLegend | Cat# 634502 |
| Rabbit anti-IST1 | Proteintech | Cat# 51002-1-AP |
| Rabbit anti-CHMP2A | Proteintech | Cat# 10477-1-AP |
| Rabbit anti-ALG2 | Proteintech | Cat# 12303-1-AP |
| Rabbit anti-Gal3 | Cell Signaling | Cat# 87985 |
| Rabbit anti-ATG12 | Cell Signaling | Cat# 2010 |
| Rabbit anti-LC3B | Cell Signaling | Cat# 2775 |
| Rabbit anti-GABARAPL1 | Cell Signaling | Cat# 26632 |
| Rabbit anti-GABARAPL2 | Cell Signaling | Cat# 14256 |

| Reagent/resource | Reference or source | Identifier or catalog number |
|---|---|---|
| Rabbit anti-ATG4B | Cell Signaling | Cat# 5299 |
| Mouse anti-β-Actin | MERCK (Sigma-Aldrich) | Cat# A2228 |
| Mouse anti-HA | ThermoFisher | Cat# 26183 |
| **Oligonucleotides and other sequence-based reagents** | | |
| PCR primers | Eurofins | Methods |
| **Chemicals, enzymes and other reagents** | | |
| Leu-Leu methyl ester hydrobromide (LLOMe) | MERCK (Sigma-Aldrich) | Cat# L7393 |
| Lysotracker Red DND-99 | ThermoFisher | Cat# L7528 |
| BAPTA-AM | ThermoFisher | Cat# B1205 |
| DAPI | ThermoFisher | Cat# 62248 |
| X-tremeGENE HP | MERCK (Sigma-Aldrich) | Cat# XTGHP-RO |
| DMEM | ThermoFisher | Cat# 41965 |
| DMEM (phenol red free) | MERCK (Sigma-Aldrich) | Cat# D1145 |
| FBS | ThermoFisher | Cat# A5256801 |
| Donkey Serum | MERCK (Sigma-Aldrich) | Cat# D9663 |
| ProLong Diamond Antifade Mountant | ThermoFisher | Cat# P36961 |
| LookOut mycoplasma PCR detection kit | MERCK (Sigma-Aldrich) | Cat# MP0035 |
| Penicillin-Streptomycin | ThermoFisher | Cat# 15140122 |
| MEM Non-Essential Amino Acids | ThermoFisher | Cat# 11140050 |
| HEPES | ThermoFisher | Cat# 15630080 |
| Opti-MEM™ Reduced Serum Medium | ThermoFisher | Cat# 31985062 |
| Trans-Blot Turbo Midi 0.2 μm Nitrocellulose Transfer Packs | Bio-Rad | Cat# 1704159 |
| Clarity Western ECL Substrate | Bio-Rad | Cat# 1705061 |
| Phusion High-Fidelity PCR Master Mix | ThermoFisher | Cat# F531 |
| **Software** | | |
| Fiji/ImageJ | v2.0 | |
| GraphPad Prism | v10.0 | |
| CellProfiler | v4.2.1 | |
| Leica LAS X Software | | |
| Imaris | v9.8 | |
| **Other** | | |
| Leica SP8 FALCON | Leica | |
| Elyra 7 lattice SIM microscope | Zeiss | |
| ChemiDoc MP Imaging System | Bio-Rad | |
| Trans-Blot Turbo transfer system | Bio-Rad | |

## Cells and cell culture

HeLa WT/ATG16L1 KO/ATG5 KO/ATG7 KO cells were a kind gift from Tomatsu Yoshimori—Osaka University, Osaka, Japan

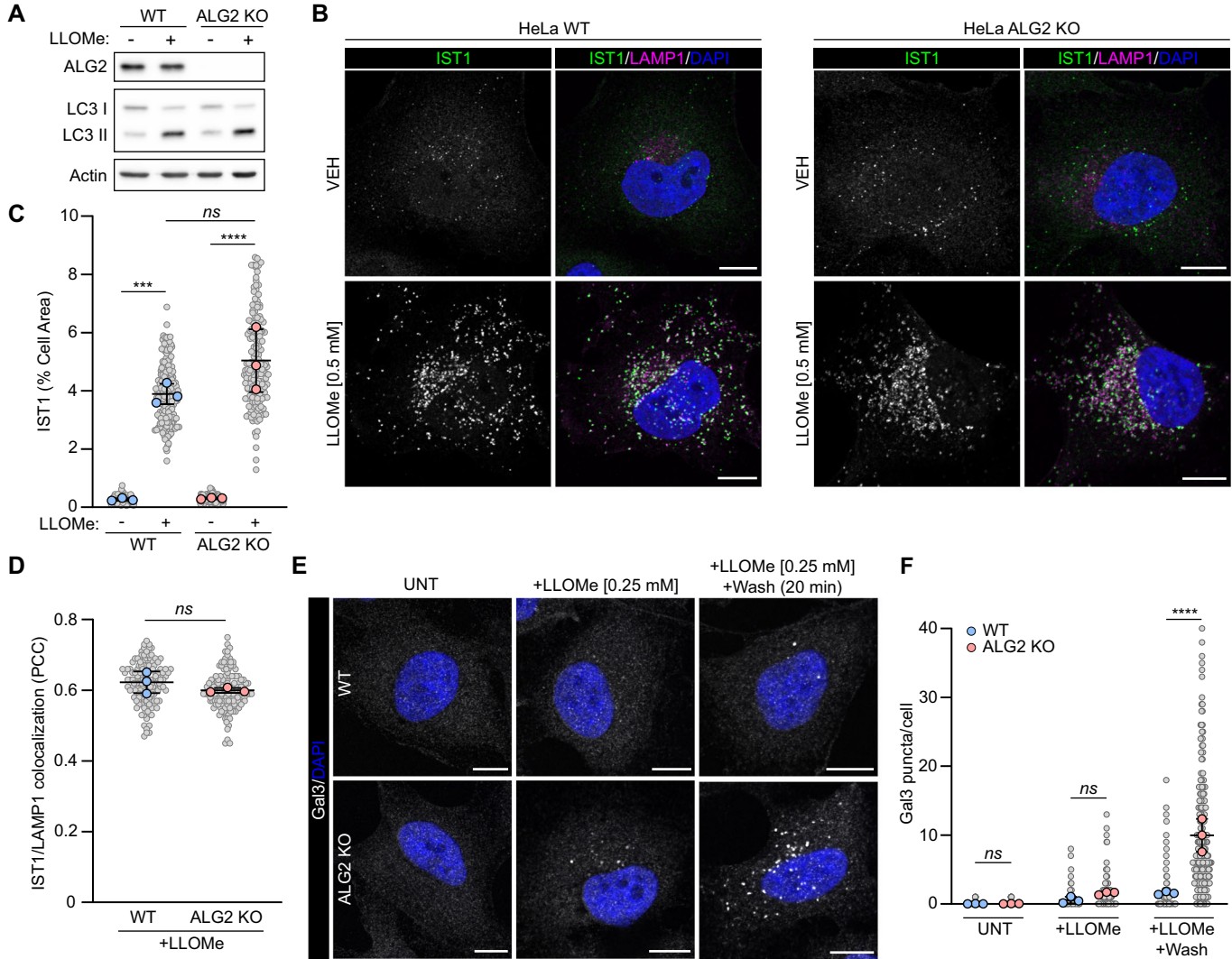

**Figure 6. ALG-2 is dispensable for ESCRT recruitment to damaged membranes, but required for efficient repair.**

(A) Western blot analysis of HeLa WT and ALG2 KO cells treated with 0.5 mM LLOMe for 15 min. (B) Representative confocal images of HeLa WT and ALG2 KO cells treated with or without 0.5 mM LLOMe for 15 min and immunostained for IST1 and LAMP1. Scale bars = 10 μm. (C) Quantification of IST1 cell area from (B). Small points represent individual cells from three independent experiments. Large points represent the means of individual experiments ($n \geq 45$ cells per experiment). Bars represent the mean ± SD from the three experiments. Significance was determined from biological replicates using a one-way ANOVA with Tukey's multiple comparisons tests. ns (not significant) ($P = 0.1393$), ***$P = 0.0002$, ****$P < 0.0001$. (D) Quantification of IST1/LAMP1 colocalization from (B). Small points represent individual cells from three independent experiments. Large points represent the means of individual experiments ($n = 45$ cells per experiment). Bars represent the mean ± SD from the three experiments. Significance was determined from biological replicates using a Student's $t$ test. ns (not significant) represents $P > 0.05$. (E) Representative confocal images of HeLa WT and ALG2 KO cells treated with or without 0.25 mM LLOMe for 5 min and immunostained for Gal3. Scale bars = 10 μm. (F) Quantification of Gal3 puncta from (B). Small points represent individual cells from three independent experiments. Large points represent the means of individual experiments ($n \geq 72$ cells per experiment). Bars represent the mean ± SD from the three experiments. Significance was determined from biological replicates using a one-way ANOVA with Tukey's multiple comparisons tests. ns (not significant) (UNT, $P > 0.9999$; +LLOMe, $P = 0.8105$), ****$P < 0.0001$. Source data are available online for this figure.

(Nakamura et al, 2020) and were characterized in our lab previously (Jia et al, 2022). HeLa WT/ATG8 KO/ATG4 KO cells were a kind gift from Michael Lazarou—Monash University, Melbourne, Australia (Nguyen et al, 2016; Nguyen et al, 2021). HeLa TECPR1 KO/E3-DKO cells were generated by CRISPR/Cas9-mediated knockout, as described below. HeLa WT/ALG2 KO cells were a kind gift from Masatoshi Maki – Nagoya University, Nagoya, Japan (Takahara et al, 2017). HEK293 WT/ATG16L1 KO were a kind gift from Anne

Simonsen—University of Oslo, Oslo, Norway (Lystad et al, 2019). HEK293 TECPR1 KO/E3-DKO cells were described previously (Corkery et al, 2023). All cells were cultured in Dulbecco's modified Eagle medium (DMEM) (Sigma-Aldrich) supplemented with 10% fetal bovine serum (FBS), 1% penicillin/streptomycin, and non-essential amino acids at 37 °C with 5% $CO_2$. Cells were routinely tested for mycoplasma contamination using the LookOut mycoplasma PCR detection kit (Sigma-Aldrich).

## Generation of CRISPR KO cell lines

Oligonucleotides encoding a gRNA targeting exon 3 of TECPR1 (CACGTAGACCTGGTTGTCAC) were annealed and cloned into pSpCas9(BB)-2A-Puro (PX459) V2.0, which expresses both the Cas9 enzyme and gRNA. HeLa WT and ATG16L1 KO cells were transiently transfected, selected with puromycin for 48 h, and clonal cell lines isolated by limiting dilution. TECPR1 KO cells were verified by genomic PCR amplification and sequencing of TECPR1 exon 3.

## Antibodies and reagents

Antibodies used in this study were from the following sources: ALIX (634502, IF 1:100) was purchased from BioLegend. IST1 (51002-1-AP, IF 1:200), CHMP2A (10477-1-AP, IF 1:100) and ALG-2 (12303-1-AP, IF 1:50) were purchased from Proteintech. Gal3 (87985, IF 1:400), ATG12 (2010, WB 1:1000), LC3B (#2775, WB: 1:1000), GABARAPL1 (#26632, WB: 1:1000), GABARAPL2 (14256, WB: 1:1000) and ATG4B (5299, WB 1:1000) were purchased from Cell Signaling. Beta-actin (A2228, WB: 1:10,000) was purchased from Sigma-Aldrich. HA Tag (26183, WB 1:5000, IF 1:200) was purchased from Thermo Fisher. Alexa Fluor 488/568/ 647 conjugated secondary antibodies for immunofluorescence were purchased from Thermo Fisher.

Reagents used in this study were from the following sources: Leu-Leu methyl ester hydrobromide (LLOMe, L7393) from Sigma-Aldrich. Lysotracker Red DND-99 (L7528), BAPTA-AM (B1205) and DAPI (62248) from Thermo Fisher.

## Immunoblotting

Cells were scraped from the plate in cold lysis buffer (20 mM Tris-HCl pH 8.0, 300 mM KCl, 10% Glycerol, 0.25% Nonidet P-40, 0.5 mM EDTA, 1 mM PMSF, 1× complete protease inhibitor (Roche)), passed through a 21 G needle and cleared by centrifugation (20 min/18,213 × $g$/4 °C). Lysates were subjected to SDS-PAGE and transferred to a 0.2 μm nitrocellulose membrane (Bio-Rad) using a Trans-Blot Turbo transfer system (Bio-Rad). Membranes were blocked in 5% skim milk (in TBST) and incubated with primary antibody diluted in 5% BSA (in TBST) overnight at 4 °C. HRP-conjugated secondary antibodies were diluted in 5% skim milk (in TBST) and incubated with the membrane for 1 h at room temperature. Protein detection was carried out using chemiluminescence (Bio-Rad) and imaged using a ChemiDoc imaging system (Bio-Rad).

## Plasmids

EGFP-TECPR1 was a kind gift from Thomas Wollert – Institute Pasteur, Paris, France (Wetzel et al, 2020). EGFP-TECPR1$^{\Delta 1-377}$ was described previously (Corkery et al, 2023). EGFP-TECPR1$^{\Delta AIR}$ was derived from EGFP-TECPR1 using PCR mutagenesis (fwd: aagaccggggcgctgcagtg, rev: catgtgtaccgaggaggacaggcc). HA-TECPR1$^{WT/\Delta 1-170/\Delta AIR}$ were generated by PCR amplifying TECPR1 $^{WT/\Delta 1-170/\Delta AIR}$ from EGFP-TECPR1 plasmids inserting an HA tag in the forward primer. LAMP1-mCherry was described previously (Corkery et al, 2023). IST1-EGFP was generated by PCR amplifying IST1 from HA-IST1 (Addgene plasmid # 131619;

https://www.addgene.org/131619/; RRID:Addgene_131619) and subcloning into the EGFP-N1 (Clontech) plasmid using NheI/SalI restriction sites. GFP-P4M-SidM was a gift from Tamas Balla (Addgene plasmid # 51469; https://www.addgene.org/51469/; RRID:Addgene_51469) (Hammond et al, 2014). pGABARAPL2 was generated by PCR amplifying human GATE16 from a cDNA library and subcloning into a mammalian expression plasmid. EGFP-Rab5 was generated by PCR amplifying Rab5a from a human cDNA library and subcloning into the pEGFPC2 vector (Clonetech) using XhoI/BamHI restriction sites. EGFP-Rab5$^{Q79L}$ was generated by PCR mutagenesis of EGFP-Rab5. mCherry-ATG5 was described previously (Corkery et al, 2023). HA-ATG5 was generated by PCR amplifying ATG5 from mCherry-ATG5, adding an HA tag in the forward primer. HA-ATG5$^{K130R}$ was derived from HA-ATG5 using PCR mutagenesis (fwd: CATTTTATGTCATG-TATGAGAGAAGCTGATGCTTTAAAAC, rev: GTTTTAAAG-CATCAGCTTCTCTCATACATGACATAAAATG). pSpCas9(BB)-2A-Puro (PX459) V2.0 was a gift from Feng Zhang (Addgene plasmid #62988; https://www.addgene.org/62988/; RRID:Addgene_62988) (Ran et al, 2013). All newly generated plasmids were verified by Sanger sequencing.

## Transfection

Transfection of DNA constructs was performed using X-tremeGENE HP transfection reagent (Sigma-Aldrich) according to the manufacturer's directions. Stable cell lines were generated via PB transposition by co-transfecting pBASE transposase with the target gene containing transposon vector at a ratio of 1:3. Cells were selected in medium containing 200 μg/mL hygromycin B for 5 days before screening for transposon integration.

## Immunofluorescence and live-cell imaging

Cells were grown on no. 1.5 glass coverslips in six-well plates. After treatment, cells were fixed in 4% paraformaldehyde for 10 min at room temperature and permeabilized in 0.25% Triton X-100 for 5 min. Cells were blocked with 5% donkey serum for 30 min followed by a 1.5 h incubation with primary antibody at room temperature. Cells incubated with Alexa Fluor conjugated secondary antibodies for 30 min at room temperature and mounted on slides using ProLong Diamond antifade mountant (Thermo Fisher).

For live-cell imaging, cells were seeded on μ-Slide eight-well slides (Ibidi) and incubated for 24 h. Imaging was performed in DMEM without phenol red (Sigma-Aldrich) and supplemented with 20 mM HEPES.

Imaging was performed on a Leica SP8 FALCON inverted confocal system (Leica Microsystems) equipped with a HC PL APO 63×/1.40 oil immersion lens and a temperature-controlled hood maintained at 37 °C and 5% $CO_2$. DAPI was excited using a 405 nm Diode laser, and EGFP/Alexa488 and mCherry/Alexa568 fluorescence were excited using a tuned white light laser. Scanning was performed in line-by-line sequential mode.

Super-resolution Images were obtained with an Elyra 7 lattice SIM microscope (Zeiss). Images were taken at ×63 with a Plan-Apochromat ×63/1.40 Oil objective using 15 phases and processed for SIM2 with the Zen Black SIM Module at default settings.

3D reconstruction of immunofluorescence images was performed using Imaris 9.8 software (Bitplane).

## LysoTrackerRED repair assay

HeLa WT/E3-DKO/ATG5 KO/ATG8 KO cells were seeded in μ-Slide eight-well glass bottom slides (Ibidi) and incubated for 24 h. Lysosomes were labelled with LysoTrackerRED (ThermoFisher) (0.75 μL in 10 mL media) for 30 min. Cells were treated with 250 μM LLOMe for 10 min, washed, and allowed to recover for 45 or 90 min in the presence of LysoTrackerRED. Cells were imaged at each time point and LysoTrackerRED area was quantified using ImageJ – FIJI distribution (NIH).

## Quantification and statistical analysis

For quantification of immunofluorescence data, 10 random fields of view were captured using a ×63 objective and ×1.5 pre-acquisition zoom (6–10 cells per image). From these images, 50–60 cells were manually extracted using ImageJ for analysis. IF images presented in the figures were captured with increased pre-acquisition zoom and are representative of the phenotype.

Batch analysis of colocalization was performed using CellProfiler 4.2.1 (Broad Institute, Inc.). Briefly, segmentation of nuclei and cell outlines was done by intensity thresholding of nuclear (DAPI) and cytoplasmic stainings (ALG2 or Gal3). Cytoplasm of individual cells were defined by subtraction of the nuclear area from cell segmentations. Colocalization of proteins (ALIX/Gal3, ALG-2/LAMP2, ALG-2/Gal3), was measured in the cytoplasm, excluding nuclear signals, and is calculated as Pearson's Correlation Coefficient (PCC). IST1/LAMP1 colocalization (Fig. 6D) was quantified using the Coloc 2 plugin in ImageJ.

Data are shown as mean ± standard deviation (SD). Statistical significance was determined by a one-way ANOVA with Tukey's multiple comparisons tests, or by Student's $t$ test, using GraphPad Prism v.10.0.0. $*P < 0.05$, $**P < 0.01$, $***P < 0.001$, $****P < 0.0001$, ns (not significant) represents $P > 0.05$. Exact $P$ values are provided in the corresponding figure legend. No blinding was done in this study.

# Data availability

This study includes no data deposited in external repositories. An earlier version of this manuscript was deposited in bioRxiv on 2024-04-30 (https://doi.org/10.1101/2024.04.30.591897).

The source data of this paper are collected in the following database record: biostudies:S-SCDT-10_1038-S44318-025-00672-1.

# Peer review information

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

## Acknowledgements

This work was supported by the European Research Council (ChemBioAP), Vetenskapsrådet (Nr. 2018-04585, Nr. 2022-02932), the Knut and Alice Wallenberg Foundation and the Göran Gustafsson Foundation for Research in Natural Sciences and Medicine to Y.W.W. We thank Irene Martinez Carrasco for assistance with the super-resolution imaging, and acknowledge the Biochemical Imaging Center at Umeå University and the National Microscopy Infrastructure, NMI (VR-RFI 2019-00217) for providing assistance in microscopy.

## Author contributions

**Dale P Corkery**: Conceptualization; Data curation; Formal analysis; Validation; Investigation; Visualization; Writing—original draft; Writing—review and editing. **Deerada Wijayatunga**: Validation; Investigation; Writing—review and editing. **Benedita K L Feron**: Validation; Investigation; Writing—review and editing. **Laura K Herzog**: Data curation; Validation; Investigation; Writing—review and editing. **Anastasia Knyazeva**: Validation; Investigation; Writing—review and editing. **Yao-Wen Wu**: Conceptualization; Formal analysis;

Supervision; Funding acquisition; Project administration; Writing—review and editing.

Source data underlying figure panels in this paper may have individual authorship assigned. Where available, figure panel/source data authorship is listed in the following database record: biostudies:S-SCDT-10_1038-S44318-025-00672-1.

## Funding

## Disclosure and competing interests statement

The authors declare no competing interests.

# Expanded View Figures

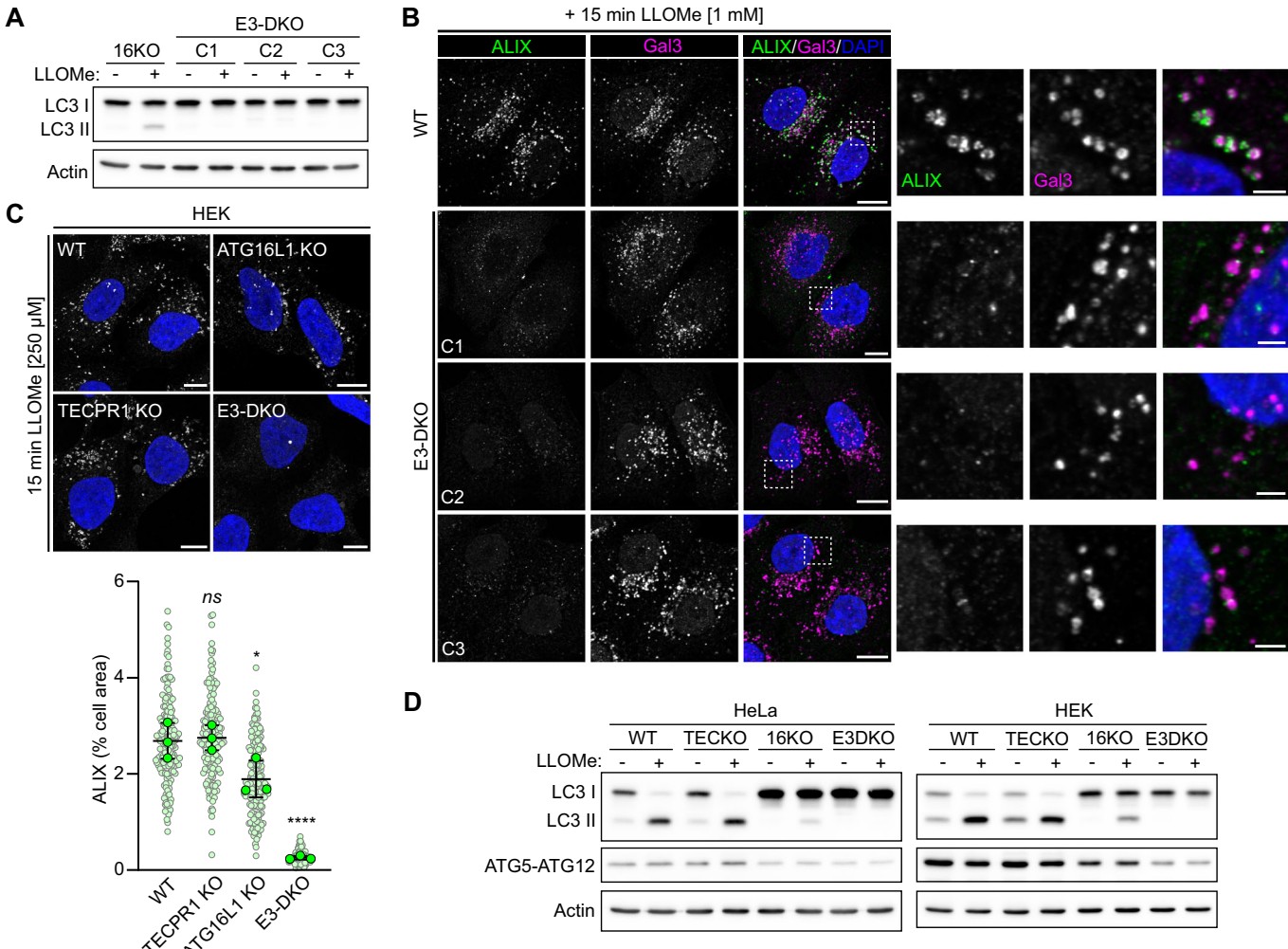

**Figure EV1.  Validation of ESCRT recruitment deficiency across multiple ATG16L1/TECPR1 DKO clones and cell lines.**

(**A**) Western blot analysis of HeLa ATG16L1 KO and three ATG16L1/TECPR1 DKO clones treated with or without 1 mM LLOMe for 30 min. (**B**) Confocal images of cell lines from (**A**) treated with 1 mM LLOMe for 15 min and immunostained for ALIX and Gal3. Nuclei were stained with DAPI. Scale bars = 10 μm for whole image and 2 μm for insets. (**C**) Top: Confocal images of HEK WT, ATG16L1 KO, TECPR1 KO and ATG16L1/TECPR1 DKO cells treated with 1 mM LLOMe for 15 min and immunostained for ALIX. Scale bars = 10 μm. Bottom: Quantification of ALIX area. Small points represent individual cells from three independent experiments. Large points represent the means of individual experiments (*n* = 60 cells per experiment). Bars represent the mean ± SD from the three experiments. Significance was determined from biological replicates using a one-way ANOVA with Tukey's multiple comparisons tests. ns (not significant) represents *P* > 0.05, **P* = 0.0314, *****P* < 0.0001. (**D**) Western blot analysis of HeLa and HEK KO cell lines treated with or without 1 mM LLOMe for 30 min. Source data are available online for this figure.

**A**

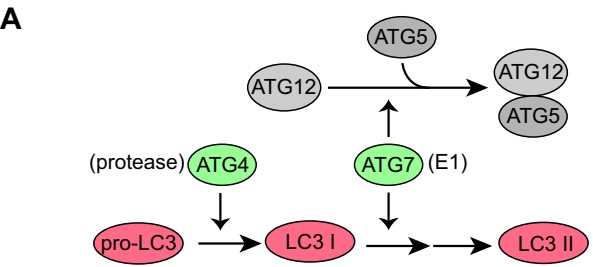

**B**

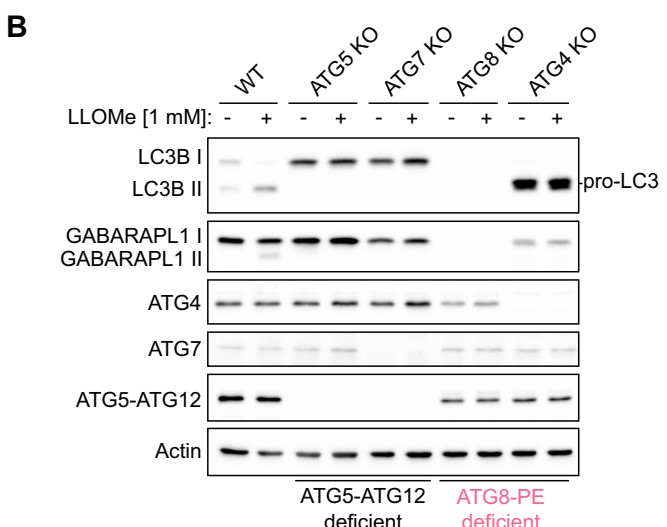

Figure EV2. **Effect of ATG KO on ATG5-ATG12 conjugation and ATG8 lipidation.**

(A) Simplified schematic of ATG8ylation pathway. (B) Western blot analysis of ATG KO cell lines. Source data are available online for this figure.

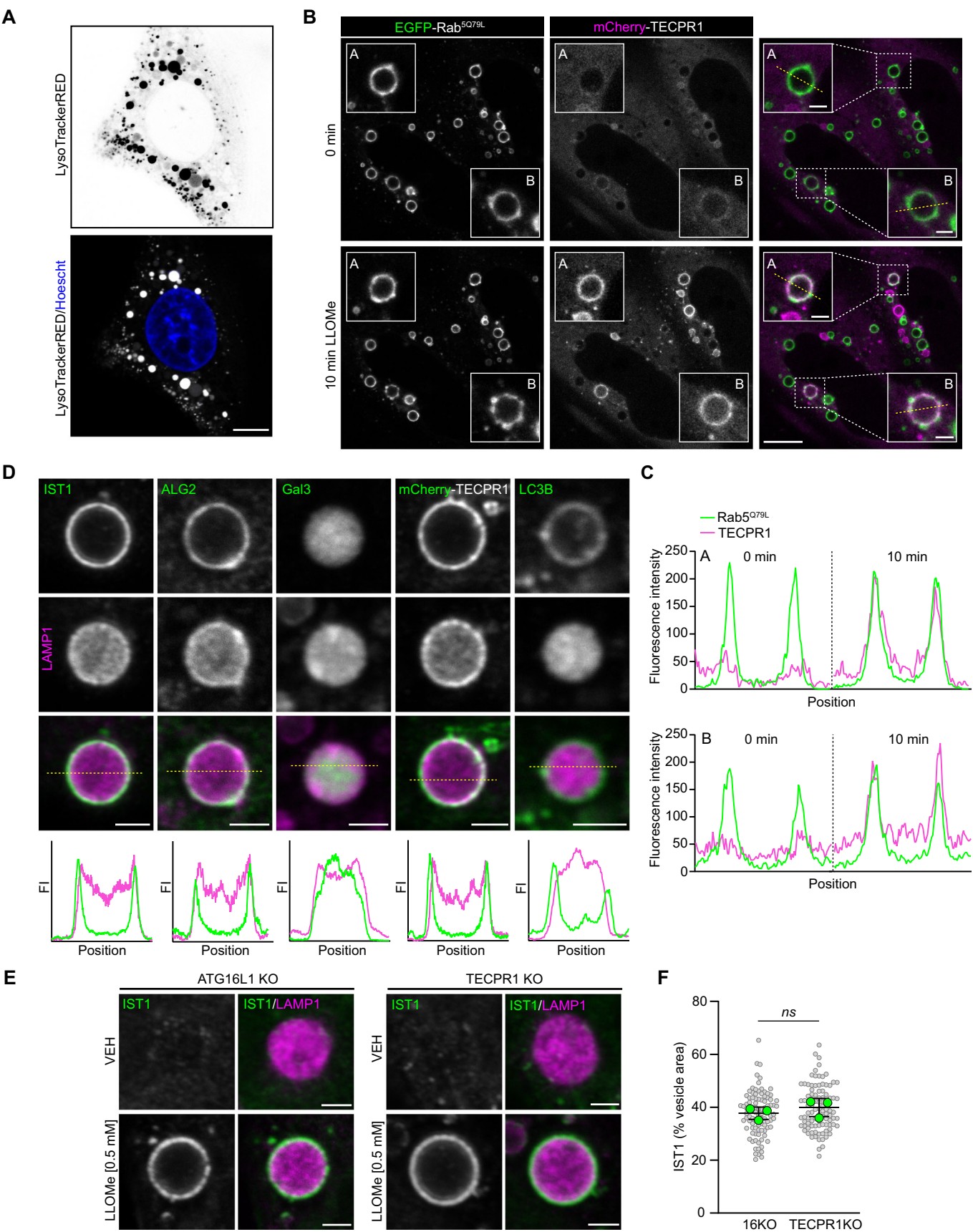

◀ **Figure EV3.   Rab5$^{Q79L}$-induced oversized vesicles can be used to model the LLOMe-induced membrane damage response.**

(A) Confocal images of a HeLa cell stably expressing Rab5$^{Q79L}$, stained with LysoTrackerRED. Nuclei were stained with Hoechst. Scale bar = 10 μm. (B) Live cell imaging of HeLa cells stable expressing EGFP-Rab5$^{Q79L}$, co-transfected with mCherry-TECPR1, and treated with 0.5 mM LLOMe for 10 min. Scale bars = 10 μm for whole cell images and 2 μm for single vesicle insets. (C) Fluorescence intensity profiles of Rab5$^{Q79L}$ and TECPR1 at oversized vesicles from (A), before and after LLOMe treatment (location marked by a yellow line on the inset image). (D) Confocal images of oversized vesicles from HeLa cells stably expressing Rab5$^{Q79L}$ treated with 0.5 mM LLOMe for 10 min. Scale bars = 2 μm. Corresponding fluorescence intensity (FI) profiles are shown below (location marked by a yellow line on the merged image). (E) Confocal images of oversized vesicles from HeLa ATG16L1 KO and TECPR1 KO cells stably expressing EGFP-Rab5Q79L treated with 0.5 mM LLOMe for 10 min. Scale bars = 2 μm. (F) Quantification of IST1 lysosomal area from (A). Grey points represent individual vesicles from three independent experiments ($n \geq 30$ vesicles per experiment). Significance was determined from biological replicates using a Student's *t* test. ns (not significant) represents $P > 0.05$. Source data are available online for this figure.

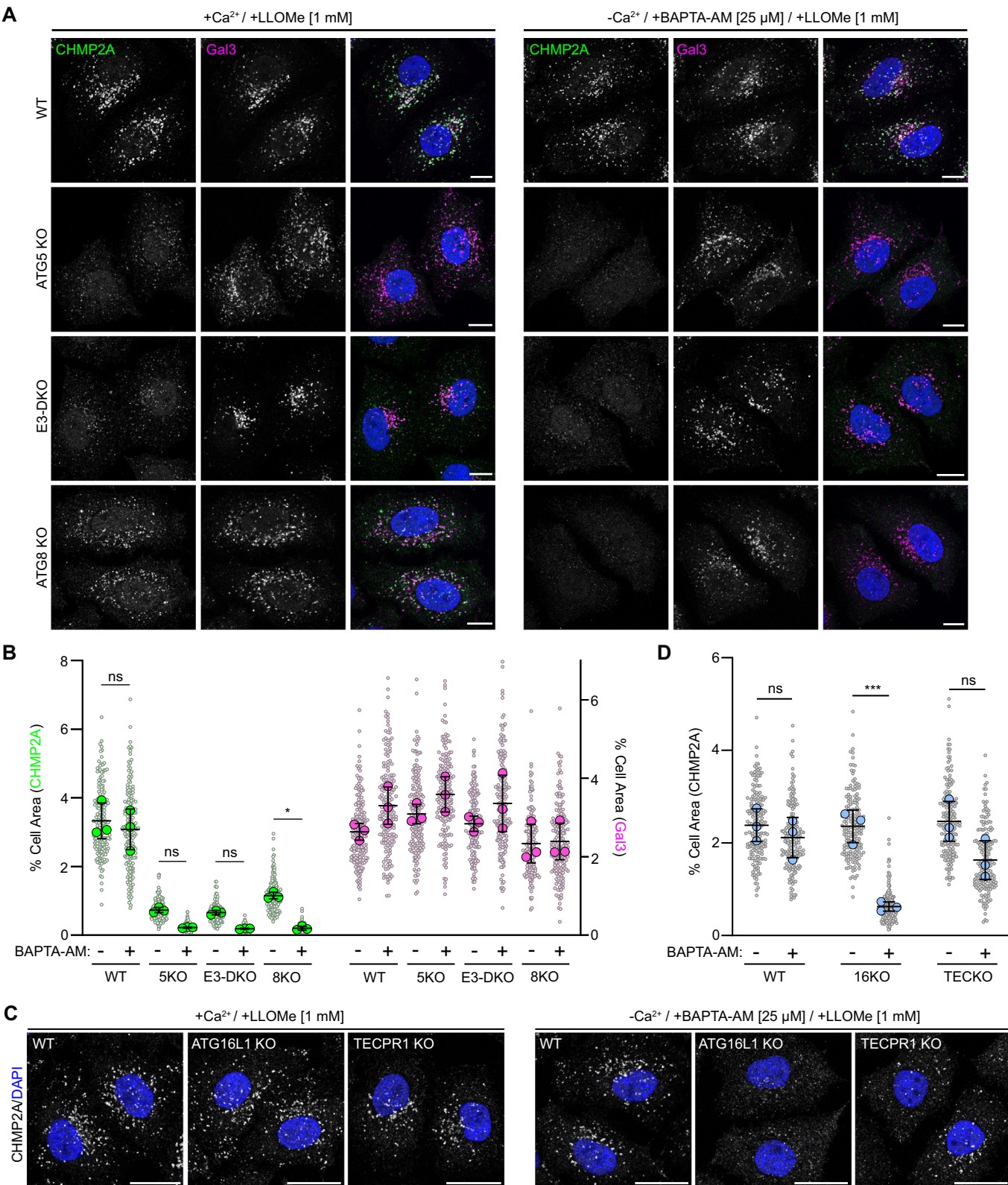

**Figure EV4. Calcium chelation inhibits ESCRT recruitment most profoundly in the absence of ATG16L1.**

(A) Confocal images of HeLa WT, ATG5 KO, E3-DKO and ATG8 KO cells treated with or without BAPTA-AM followed by 1 mM LLOMe for 30 min and immunostained for CHMP2A and Gal3. Scale bars = 10 μm. (B) Quantification of CHMP2A/Gal3 cell area from (A). Small points represent individual cells from three independent experiments. Large points represent the means of individual experiments ($n = 60$ cells per experiment). Bars represent the mean ± SD from the three experiments. Significance was determined from biological replicates using a one-way ANOVA with Tukey's multiple comparisons tests. ns (not significant) represents $P > 0.05$, $*P = 0.0134$. (C) Confocal images of HeLa WT, ATG16L1 KO and TECPR1 KO cells treated with or without BAPTA-AM followed by 1 mM LLOMe for 30 min and immunostained for CHMP2A. Scale bars = 20 μm. (D) Quantification of CHMP2A cell area from (C). Small points represent individual cells from three independent experiments. Large points represent the means of individual experiments ($n = 60$ cells per experiment). Bars represent the mean ± SD from the three experiments. Significance was determined from biological replicates using a one-way ANOVA with Tukey's multiple comparisons tests. ns (not significant) represents $P > 0.05$, $***P = 0.0009$. Source data are available online for this figure.

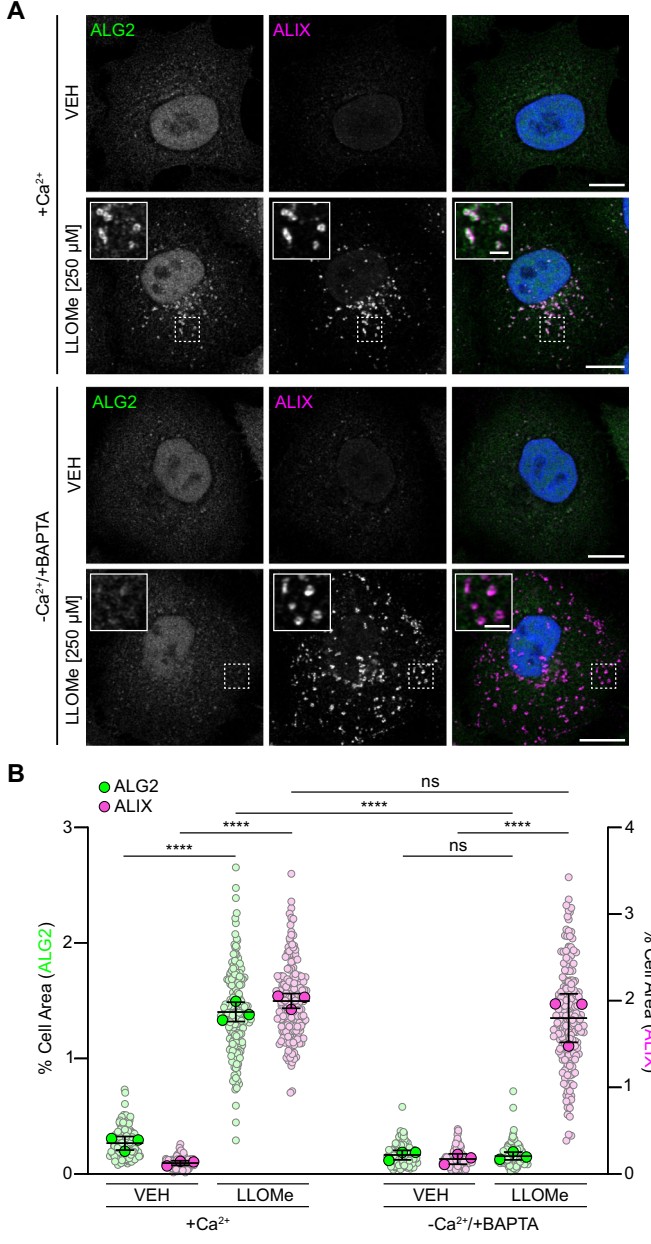

**Figure EV5.   Calcium chelation prevents ALG-2 recruitment to damaged lysosomes.**

(**A**) Confocal images of HeLa WT cells treated with or without BAPTA-AM followed by 1 mM LLOMe for 30 min and immunostained for ALG-2 and ALIX. Scale bars = 10 μm for whole cell images and 2 μm for insets. (**B**) Quantification of ALG-2/ALIX cell area from (**A**). Small points represent individual cells from three independent experiments. Large points represent the means of individual experiments (*n* = 60 cells per experiment). Bars represent the mean ± SD from the three experiments. Significance was determined from biological replicates using a one-way ANOVA with Tukey's multiple comparisons tests. ns (not significant) (LLOMe ALIX, *P* = 0.4283; BAPTA VEH/LLOMe ALG-2, *P* = 0.9981), ****$P$ < 0.0001. Source data are available online for this figure.

