## [Peer Review File · The EMBO Journal]

ATG8 E3-like ligases sense lysosomal damage and initiate ESCRT-mediated membrane repair

Dale Corkery, Deerada Wijayatunga, Benedita Feron, Laura Herzog, Anastasia Knyazeva, and Yaowen Wu

Corresponding author(s): Yaowen Wu (yaowen.wu@umu.se) , Dale Corkery (dale.corkery@umu.se)

Review Timeline:

Submission Date:	11th Jun 24
Editorial Decision:	2nd Aug 24
Appeal Received:	14th Aug 25
Editorial Decision:	15th Sep 25
Revision Received:	23rd Sep 25
Editorial Decision:	12th Nov 25
Revision Received:	13th Nov 25
Accepted:	21st Nov 25

Editor: William Teale

Transaction Report:

Dear Prof. Wu,

Thank you for submitting a revised manuscript EMBOJ-2024-118953 'ESCRT recruitment to damaged lysosomes is dependent on the ATG8 E3-like ligases'. I have now received reports from two reviewers, which I attach below. I apologise for the unusually long time that the manuscript has been with me. Your work describes a mechanism for the recruitment of ALG-2 and ESCRT complex to damaged lysosomes, describing both LC3B-dependent (via an LIR motif in calcium-activated ALG-2) and LC3B-independent (via direct interaction with ATG5 and ATG12) protein-protein interfaces. These are findings which would be of interest to our readership.

However, in their reports, both reviewers asked for more mechanistic resolution on the contributions of these modules, especially in the context of different types of membrane damage and wider lysosome-related processes. I therefore cannot, at present, offer publication in EMBO Journal. I would, however, be very happy to discuss these reports with you to explore whether the referees' concerns can be addressed and we can find a path to publication. Please let me know a convenient time, and we can discuss over Zoom next week.

Yours sincerely,

William Teale

William Teale, PhD
Editor
The EMBO Journal
w.teale@embojournal.org

Referee #1:

The manuscript by Corkery and colleagues explores a novel mechanistic role of the ATG8 E3-like ligases, specifically the ATG5-ATG12 complex, in the recruitment of the ESCRT machinery to damaged lysosomes. This process is mediated by ALG-2 in response to calcium (Ca²⁺) leakage following lysosomal membrane damage. The data presented are comprehensive and thorough, addressing key mechanisms involved. However, this study builds heavily on previous work from the authors and other colleagues, and has an incremental added value. Furthermore, certain results require better definition and quantification, as specified in detail below. Most notably, the authors never show (starting in Figure 1) that the different proteins are colocalizing with lysosomes - only with Gal3, which is indeed expected to be at the lysosomal damage sites, but a technical demonstration would be in order. The imaging data is presented in very small format, and the visualization of possible colocalizations (crucial for the interpretation of the data in this study) is often difficult or impossible (even with the images magnified in the computer). Insets with magnification are important to facilitate the reading of the figures.

Several crucial measurements are missing, such as the colocalization of various proteins (e.g., Alix and Gal3 in Figure 1, CHMP2A and LAMP1/Gal3). It is also unclear why only the double knockout (KO) of ATG16L1 and TECPR1 affected ALIX/ALG-2 recruitment, whereas the single KOs did not, despite the authors' claim that these two proteins have redundant functions. Given their different targets, this explanation seems insufficient. The proposed kinetics of protein recruitment in this study have some weaknesses, particularly in the presentation and discussion of the results. The authors should consider including live imaging experiments to support their hypothesis: evaluating the kinetic of ATG5 acquisition of damaged lysosomes and ALG-2 and/or ATG16L1/TECPR1. Additionally, the validation of the main findings in a physiological model was not presented.

1. The authors should clarify that ATG16L1 is recruited to lysosomes and interacts with V-ATPase even under basal conditions.
2. Figure 1A: The representative images of Gal3 puncta do not match the results shown in Figure 1B. Additionally, the authors should provide cropped images from each panel, similar to those in Figure 1C.
3. Figure 1C: The Gal3 puncta appear much stronger than in control cells, which is inconsistent with Figures 1A and B. This discrepancy raises questions about whether ATG16L1, TECPR1, and E3-DKO cells have inherent endocytic deficiencies, leading to different basal levels of lysosomal damage. Furthermore, considering the impairment in lysosomal membrane repair, shouldn't the basal levels after LLOMe treatment be increased? Minor points: The scale bars are inconsistent across different panels, and the channels should be presented individually.
4. Figure 1G: The authors should provide cropped images to better illustrate the results.
5. Figure 2B: The western blot should be performed in the presence of LLOMe and could be included as supplementary information.
6. Figure 2F: The authors should present split channels, and again, some representative images do not reflect the quantification.

In ATG8KO cells, the ALIX levels appear much lower than Gal3, even though in the cropped image, the ALIX staining seems more intense than in the original image. As mentioned before, it would be more accurate to calculate the colocalization between Gal3 and ALIX.

7. Figure 2F and other panels: The method for selecting KO cells for imaging and quantification is not explained.
8. Figures 3 and 4: The authors should maintain the use of Gal3-positive vesicles as a readout for damaged lysosomes. Otherwise, without showing the cellular response in control/basal conditions, the results are weaker.
9. Figure 3D: Minor point - please ensure proper staining labels.
10. Figure 3E: A 10-minute LLOMe treatment induces only a small percentage of lysosomal damage. Therefore, the results presented may appear exaggerated compared to what is typically observed for such a short treatment duration. The authors should present the Gal3 puncta analysis over the time course used in this condition.
11. Figures 3C and D: Both figures present similar results. Consider moving one panel to the supplementary information.
12. Figure 4: Quantification of images (A and C) is missing. The authors should use Gal3 as a marker of lysosomal damage rather than labeling lysosomes with LAMP1. Subsequently, colocalization between ALG-2 and Gal3 should be performed under different conditions.
13. Figures 4/5: The authors should quantify the levels of CHMP2A and ALG-2 recruitment and interaction in conditions of lysosomal damage, with and without a calcium chelator (BAPTA), in control cells, ATG5-KO, and ATG8-KO. Additionally, ATG5 recruitment to damaged lysosomes should be assessed in the presence of BAPTA.
14. Minor points: Page 6, line 184: Specify the cell lines used. Line 188: Define the mutant ATG5.
15. The recruitment of the ALG-2 LIR mutant to damaged lysosomes should be addressed.
16. Minor points: Several cropped images lack scale bars, and the scale value is not mentioned in the figure legends. The line used to measure intensity for histograms should also be shown.
17. The knockout of several proteins was not validated in this study, and either references or validation should be presented.

Referee #2:

The ESCRT machinery is crucial for repairing damaged endolysosomal membranes. However, how cells detect this damage and recruit ESCRT is not well understood. The current study by Corkery et al reveals that the translocation of autophagy-related ATG8 E3-like ligases to lysosomal membranes upon damage acts as a catalyst for ESCRT recruitment. The authors propose that leakage of protons or calcium from lysosomes triggers this recruitment through different pathways: V-ATPase-dependent for protons and sphingomyelin-dependent for calcium. The ATG16L1-ATG5-ATG12 or TECPR1-ATG5-ATG12 E3-like complexes are necessary for this process. The ATG5-ATG12 complex stabilizes the calcium sensor ALG-2 and recruits the repair complex, with LC3B binding directly to Ca²⁺-activated ALG-2. Thus, ATG8 E3-like ligases serve as damage sensors for ESCRT-mediated membrane repair.

Overall this is an interesting manuscript putting forward a new hypothesis on how lysosomal membrane damage is linked to ESCRT-III recruitment, an open question in the field. However, there are notable weaknesses in the proposed mechanism. In particular, the authors build on a model where ESCRT-III recruitment during lysosomal damage is thought to be mediated by the calcium-binding protein, ALG-2. However, this model is increasingly recognized to be damage type-specific, and likely does not apply to the specific perturbation applied here (LLOMe). A key piece of data is missing, where the requirement for ALG2 and its interaction with LC3 in ESCRT-III recruitment is functionally tested (see point 3 below).

Main concerns:

- 1- With regards to uniform vs fragmented distribution of ESCRT-III components upon ATG5-KO and ATG8-KO (Fig. 3C, 3D) - what is the distribution of ATG5 and ATG8 on lysosomes upon LLOMe treatment? Is it also uniform or are there certain locations where these proteins localize more? Does the uniformity of the distribution change upon single KO of TECPR1 or ATG16L1?
- 2- In the LysoTrackerRED LLOMe-washout experiment (Fig. 3E, 3F), how do the authors differentiate membrane repair with lysosome biogenesis or other vesicle trafficking processes? A better experimental set-up is to pre-load lysosomes with AF488 and AF568-labeled Dextran and repeat the same experimental paradigm to better differentiate repair of pre-existing lysosomes with lysosome biogenesis. This is an important distinction since ATG8-KO may also impair lysophagy processes.
- 3- Regarding recruitment of ALG-2 in an LIR-dependent manner to lysosomes - the authors are missing a few key data:
 - Do ALG-2 LIR mutants fail to get recruited to damaged lysosomes (by IF)? Authors have only done pull-down experiments with these mutants, and only in the context of Ca²⁺ addition, but not LLOMe addition.
 - Do ALG-2 LIR mutants fail to recruit ESCRT-III components upon lysosome damage by LLOMe? KD-rescue experiments are needed to substantiate this key point.Of note, in the Chen et al, 2024 (PMID: 38781205) manuscript, the authors found that ALG-2 dependent ESCRT-III recruitment was only required for GPN-dependent lysosome damage, but not LLOMe-dependent lysosome damage. Do the authors find the same in their system, and can they comment on the differences/similarities?

Additional concerns on specific data:

- Figure 1B: It would be clearer if the authors show statistical comparisons between marker between genotype, not between marker within a genotype- as the conclusion being made is that the E3 DKO has less ALIX recruitment compared to WT but not the single KOs.
- Figure 1B: Why does it appear that the E3 DKO is not more damaged based on Gal3 recruitment? Have the authors done a time course to see whether the E3 DKO recruits Gal3 more quickly or accumulates more Gal3 over time?
- Figure 3B: Is it that less ESCRT-III is recruited upon ATG8 KO or just that it is recruited slower?
- Figure 4A: The authors should quantify ALG-2 on lysosomes, as the differences will be clearer and more convincing.
- In Figure 5A, the differences do not appear to be drastic by eye. The authors should do further in cell experiments: in LC3 WT or F52A expressing cells +/- LLOMe, whether ALG-2 recruitment is impaired, and likewise verify ALG-2 WT or F32A mutant recruitment +/- LLOMe.

** As a service to authors, EMBO Press provides authors with the possibility to transfer a manuscript that one journal cannot offer to publish to another EMBO publication or the open access journal Life Science Alliance launched in partnership between EMBO Press, Rockefeller University Press and Cold Spring Harbor Laboratory Press. The full manuscript and if applicable, reviewers' reports, are automatically sent to the receiving journal to allow for fast handling and a prompt decision on your manuscript. For more details of this service, and to transfer your manuscript please click on Link Not Available. **

Dear editors and reviewers,

We would like to thank you for giving us the opportunity to submit a revised version of our manuscript. We would also like to express our gratitude to the editors and the reviewers for the time and effort that you dedicated to providing valuable feedback on our manuscript. We are also pleased that you and the reviewers appreciate the importance of our study. We are grateful to the reviewers for their insightful comments, which have contributed to this improved version of our manuscript. We have carefully addressed the expressed concerns and incorporated the suggestions provided by the reviewers. All changes have been highlighted within the manuscript.

Below is a point-by-point response to the reviewers' comments and concerns.

Referee #1:

The manuscript by Corkery and colleagues explores a novel mechanistic role of the ATG8 E3-like ligases, specifically the ATG5-ATG12 complex, in the recruitment of the ESCRT machinery to damaged lysosomes. This process is mediated by ALG-2 in response to calcium (Ca²⁺) leakage following lysosomal membrane damage. The data presented are comprehensive and thorough, addressing key mechanisms involved. However, this study builds heavily on previous work from the authors and other colleagues, and has an incremental added value. Furthermore, certain results require better definition and quantification, as specified in detail below. Most notably, the authors never show (starting in Figure 1) that the different proteins are colocalizing with lysosomes - only with Gal3, which is indeed expected to be at the lysosomal damage sites, but a technical demonstration would be in order. The imaging data is presented in very small format, and the visualization of possible colocalizations (crucial for the interpretation of the data in this study) is often difficult or impossible (even with the images magnified in the computer). Insets with magnification are important to facilitate the reading of the figures.

Several crucial measurements are missing, such as the colocalization of various proteins (e.g., Alix and Gal3 in Figure 1, CHMP2A and LAMP1/Gal3). It is also unclear why only the double knockout (KO) of ATG16L1 and TECPR1 affected ALIX/ALG-2 recruitment, whereas the single KOs did not, despite the authors' claim that these two proteins have redundant functions. Given their different targets, this explanation seems insufficient. The proposed kinetics of protein recruitment in this study have some weaknesses, particularly in the presentation and discussion of the results. The authors should consider including live imaging experiments to support their hypothesis: evaluating the kinetic of ATG5 acquisition of damaged lysosomes and ALG-2 and/or ATG16L1/TECPR1. Additionally, the validation of the main findings in a physiological model was not presented.

We would like to start by thanking the reviewer for their constructive comments. In the revised manuscript, we have made a concerted effort to improve the presentation of imaging data and added additional colocalization data to aid in interpretation (addressed in the responses below). Regarding the use of Gal3 as a marker of lysosome damage, we feel it is important to note that the cellular response to membrane damage involves two divergent pathways activated according to the extent of the damage. Moderate membrane damage engages membrane repair pathways, in which the ESCRT machinery plays a predominant role. Extensively damaged/ruptured membranes engage the membrane removal pathway, for which, galectin recruitment is believed to be the initiating signal (PMID: 22246324, 30314966). As we, and others, have shown that the recruitment of the ATG8 E3-like ligases and ESCRT machinery precedes Gal3 recruitment, colocalization of Gal3 with ESCRT/E3 ligases is largely dose/time dependent, and should be interpreted with caution. Inclusion of Gal3 in Figures 1 and 2 was intended to demonstrate that

LLOMe treatment was indeed inducing membrane damage in the E3 DKO cell line, as the complete absence of ESCRT recruitment seemed remarkable.

Regarding the functional redundancy concerns: while it is clear that TECPR1 and ATG16L1 are indeed recruited to damaged membranes through different targets (sphingomyelin vs. V-ATPase), the functional consequence of their recruitment remains the same (ie. ATG5-ATG12 recruitment and membrane ATG8ylation). As we demonstrate that ATG5-ATG12 and membrane ATG8ylation are responsible for ESCRT recruitment, we believe our explanation of functional redundancy is both logical and sufficient.

Live cell and time-course imaging experiments have been used to evaluate the kinetic recruitment of ESCRT components and Gal3 in WT and E3-DKO cells (Figure 1E, EV2).

1. The authors should clarify that ATG16L1 is recruited to lysosomes and interacts with V-ATPase even under basal conditions.

Re1: Recent reports have shown that ATG16L1 interacts with the V₁H subunit of fully assembled V-ATPase (PMID: 39089251), linking membrane damage induced V-ATPase assembly to ATG16L1 recruitment. Interestingly, the authors observed that ATG16L1 preferentially binds inactive, intact V-ATPase complexes suggesting this interaction may be further regulated by V-ATPase inactivation, which occurs at damaged membranes after attempts to restore the proton gradient have failed. Thus, whether or not V-ATPase and ATG16L1 interact under basal conditions has yet to be fully established, and is not something we can comment on in this manuscript.

2. Figure 1A: The representative images of Gal3 puncta do not match the results shown in Figure 1B. Additionally, the authors should provide cropped images from each panel, similar to those in Figure 1C.

Re2: While we do observe significant variability in Gal3 staining across a cell population (as is reflected in the quantification), we have replaced the representative image for ATG16L1 KO to better reflect the mean response. We have also re-analyzed the data set for ALIX/Gal3 colocalization, which is now presented as Figure 1C. Cropped images for each panel have also been added to Figure 1A.

3. Figure 1C: The Gal3 puncta appear much stronger than in control cells, which is inconsistent with Figures 1A and B. This discrepancy raises questions about whether ATG16L1, TECPR1, and E3-DKO cells have inherent endocytic deficiencies, leading to different basal levels of lysosomal damage. Furthermore, considering the impairment in lysosomal membrane repair, shouldn't the basal levels after LLOMe treatment be increased? Minor points: The scale bars are inconsistent across different panels, and the channels should be presented individually.

Re3: We have adjusted Figure 1C to present ALIX and Gal3 staining as separate channels. In this format, it is clear that the Gal3 levels do not appear markedly different between WT and E3-DKO cells (now Figure 1D).

The reviewer's point regarding basal levels of damage is a good one. If we use Gal3 recruitment as an indicator of membrane rupture, we would anticipate that E3-DKO cells are more susceptible to damage and, therefore, progress to rupture more quickly. A 30-minute treatment with 1 mM LLOMe is quite harsh and would be sufficient to severely damage the whole lysosome population, so changes in dynamics would not be obvious under the conditions used for these experiments. To address this point, we performed a time course experiment to assess ALIX and Gal3 recruitment in WT and E3-DKO cells treated with 250 μ M LLOMe (**Figure R1, new Figure EV2**). Untreated cells do not show any basal levels of membrane damage (ALIX/Gal3) likely due to

parallel ESCRT-independent membrane repair pathways capable of repairing endogenous damage (PMID: 35388011, 38348092, 36071159). At later time points (15/20 minutes) we do observe a significant increase in Gal3 staining in E3-DKO cells, relative to WT. This data suggests that the inability to recruit ESCRT machinery increases lysosomal susceptibility to rupture. This point is explored further in **Figure R11, new Figure 4A**.

Figure R1 (new data; Figure EV2). E3-DKO cells are more susceptible to lysosome rupture. (A) Confocal images of HeLa WT and E3-DKO cells treated with 250 μ M LLOMe for the indicated time period. Scale bars = 10 μ m. (B) Quantification of ALIX and Gal3 area from (A). Small points represent individual cells from three independent experiments. Large points represent the means of individual experiments ($n = 60$ cells per experiment). Bars represent the mean \pm SD from the three experiments. Significance was determined from biological replicates using a one-way ANOVA with Tukey's multiple comparisons tests. ns = not significant, * $p = 0.0459$, ** $p = 0.0050$, ** $p < 0.0001$.**

Regarding the scale bar inconsistency: all imaging was performed with a 63x objective, but pre-acquisition zoom was applied to retain pixel density of magnified images. Inconsistency in scale bars represent the slight differences in zoom between images.

4. Figure 1G: The authors should provide cropped images to better illustrate the results.

Re4: Magnified insets have been added (now Figure 1H).

5. Figure 2B: The western blot should be performed in the presence of LLOMe and could be included as supplementary information.

Re5: The western blot has been repeated with lysates from all cell lines treated with and without LLOMe (**Figure R2, new Figure EV5B**) and panels A and B have been moved to Figure EV5 in the revised manuscript.

FIGURE R2 (new data; Figure EV5B). Western blot analysis of HeLa KO cell lines treated with or without 1 mM LLOMe for 30 minutes.

6. Figure 2F: The authors should present split channels, and again, some representative images do not reflect the quantification. In ATG8KO cells, the ALIX levels appear much lower than Gal3, even though in the cropped image, the ALIX staining seems more intense than in the original image. As mentioned before, it would be more accurate to calculate the colocalization between Gal3 and ALIX.

Re6: We have adjusted Figure 2 to present ALIX and Gal3 staining as separate channels.

The remainder of this comment we believe is referring to Figure 2C (new Figure 2A). The enlarged images presented in this figure were not cropped from the parental image, but rather captured separately with increased pre-acquisition zoom to improve resolution/pixel density. This is why insets may appear slightly different from the parental image.

While colocalization of ALIX and Gal3 does generally follow area measurements (as shown in Figure 1 B and C), the timing of their recruitment means they may not always occupy the same lysosomal membrane during the damage response (see Figure R11, new Figure 4A). As we are interested mainly in ESCRT recruitment, and use Gal3 staining as a control to show damage has occurred, we believe area measurements are the proper quantification method in this case.

7. Figure 2F and other panels: The method for selecting KO cells for imaging and quantification is not explained.

Re7: For quantification, 10 random fields of view were captured using a 63x objective and 1.5x pre-acquisition zoom (6 – 10 cells per image). From these images, 50-60 cells were manually extracted using ImageJ for analysis. IF images presented in the figures were captured with increased pre-acquisition zoom and intended to be representative of the phenotype. These details have been added to the methods section.

8. Figures 3 and 4: The authors should maintain the use of Gal3-positive vesicles as a readout for damaged lysosomes. Otherwise, without showing the cellular response in control/basal conditions, the results are weaker.

Re8: As described above, Gal3 marks ruptured lysosomes. As Figure 3 focuses specifically on the repair pathway, we have shifted our attention to ESCRT markers exclusively. Previous figures established LLOMe dosing as being sufficient to induce damage in these cell lines, and show no staining under basal conditions.

ALG2 recruitment to the damaged lysosomes (Figure 4) has been further supported with ALG-2/Gal3 colocalization data, as requested (Figure R3, new Figure EV10).

Figure R3 (new data; Figure EV10). ALG-2 recruitment to damaged lysosomes is dependent on ATG8 and the ATG8 E3-like ligases. (A) Confocal images of HeLa WT, E3-DKO and ATG8 KO cells treated with 1 mM LLOMe for 30 minutes. (B) Quantification of ALG-2/Gal3 colocalization from (A). Small points represent individual cells from three independent experiments. Large points represent the means of individual experiments ($n > 50$ cells per experiment). Bars represent the mean \pm SD from the three experiments. Significance was determined from biological replicates using a one-way ANOVA with Tukey's multiple comparisons tests. ns = not significant, **** $p < 0.0001$.

9. Figure 3D: Minor point - please ensure proper staining labels.

Re9: Additional labels have been added for clarity.

10. Figure 3E: A 10-minute LLOMe treatment induces only a small percentage of lysosomal damage. Therefore, the results presented may appear exaggerated compared to what is typically observed for such a short treatment duration. The authors should present the Gal3 puncta analysis over the time course used in this condition.

Re10: The dose and time of LLOMe treatment was intentionally reduced in this experiment to limit the amount of damage inflicted on lysosomes. Pushing the damage to the point of rupture (Gal3 staining) will promote membrane removal rather than repair. **Figure R1** (EV2 in the manuscript) shows that a 10-minute treatment with 250 μ M LLOMe is sufficient to induce strong ESCRT recruitment, but limited Gal3 recruitment suggesting lysosomes are thoroughly damaged, but not yet ruptured.

11. Figures 3C and D: Both figures present similar results. Consider moving one panel to the supplementary information.

Re11: Figure 3C has been moved to Figure EV6 in the revised manuscript.

12. Figure 4: Quantification of images (A and C) is missing. The authors should use Gal3 as a marker of lysosomal damage rather than labeling lysosomes with LAMP1. Subsequently, colocalization between ALG-2 and Gal3 should be performed under different conditions.

Re12: Colocalization analysis has been performed for the experiment presented in Figure 4A (**Figure R4, new Figure 5B**). Due to high background of both ALG2 and HA antibodies, we were unable to perform an accurate colocalization analysis for Figure 4C (now Figure 5D). However, the recruitment of the E3-like complexes to the damaged lysosome has been well documented (PMID: 29317426, 37409490, 37409525, 37381828).

Additional colocalization analysis for ALG-2 and Gal3 in WT, E3-DKO and ATG8 KO cells treated with LLOMe has been added (**Figure R3, new Figure EV10**).

FIGURE R4 (new data; Figure 5B). Colocalization analysis of data presented in Figure 4A. Small points represent individual cells from three independent experiments. Large blue points represent the means of individual experiments ($n > 50$ cells per experiment). Bars represent the mean \pm SD from the three experiments. Significance was determined from biological replicates using a one-way ANOVA with Tukey's multiple comparisons tests. ns = not significant, ** $p = 0.0014$, *** $p = 0.0004$.

13. Figures 4/5: The authors should quantify the levels of CHMP2A and ALG-2 recruitment and interaction in conditions of lysosomal damage, with and without a calcium chelator (BAPTA), in control cells, ATG5-KO, and ATG8-KO. Additionally, ATG5 recruitment to damaged lysosomes should be assessed in the presence of BAPTA.

Re13: We thank the reviewer for this insightful suggestion. Using a calcium chelation protocol of 25 μ M BAPTA-AM for 45 minutes prior to LLOMe treatment (1 mM – 30 minutes), we still observed significant CHMP2A recruitment to damaged lysosomes in WT cells (**Figure R5A, B, new Figure EV11A, B**). This chelation protocol was, however, sufficient to prevent ALG-2 recruitment (**Figure R6, new Figure EV12**) suggesting ALG-2 is not responsible for ESCRT

recruitment to lysosomes damaged with LLOMe – a point addressed below in our response to referee #2, Re4.

With ALG-2 proving to be dispensable for ESCRT recruitment, we wondered what role calcium plays in ESCRT recruitment, and why calcium chelation failed to inhibit ESCRT recruitment in WT cells as BAPTA pre-treatment has been shown to inhibit ESCRT recruitment throughout the literature (PMID: 32643832, 31813797, 29622626). It is worth noting that the extent to which BAPTA pre-treatment inhibits ESCRT recruitment in the above referenced studies varies significantly, which could be due to timing (PMID: 31813797) or variations in chelation efficacy. An additional explanation for this variability could be due to our observation that ATG16L1 KO cells, but not TECPR1 KO cells, displayed increased susceptibility to calcium chelation (**Figure R5C, D, new Figure EV11C, D**). Knockout of ATG16L1 presumably poses increased demand on the TECPR1 containing E3-like ligase complex for ESCRT recruitment to damaged membranes. As the scrambling of sphingomyelin at damaged membranes has been shown to be calcium dependent (PMID: 35388011), calcium chelation should preferentially inhibit recruitment of the TECPR1 E3-like ligase complex, but not the ATG16L1 containing complex. We have shown that ATG16L1 can fully compensate for the loss of TECPR1 in the cells lines employed in this study, which likely explains why calcium chelation failed to inhibit ESCRT recruitment in WT cells. Variability in calcium chelation efficacy suggests that different cell lines may preferentially use one E3-like ligase complex over that other – a concept we aim to explore further.

The revised manuscript includes a detailed dissection of the role of ALG-2 and calcium in LLOMe-induced lysosomal membrane damage.

Figure R5 (new data; Figure EV11). Calcium chelation inhibits ESCRT recruitment most profoundly in the absence of ATG16L1 (A) Confocal images of HeLa WT, ATG5 KO, E3-DKO and ATG8 KO cells treated with or without BAPTA-AM followed by 1 mM LLOMe for 30 minutes and immunostained for CHMP2A and Gal3. Scale bars = 10 μm. (B) Quantification of CHMP2A/Gal3 cell area from (A). Small points represent individual cells from three independent experiments. Large points represent the means of individual experiments (n = 60 cells per experiment). Bars represent the mean ± SD from the three experiments. Significance was determined from biological replicates using a one-way ANOVA with Tukey's multiple comparisons tests. ns = not significant, * p = 0.0134. (C) Confocal images of HeLa WT, ATG16L1 KO and TECPR1 KO cells treated with or without BAPTA-AM followed by 1 mM LLOMe for 30 minutes and immunostained for CHMP2A. Scale bars = 20 μm. (D) Quantification of CHMP2A cell area from (C). Small points represent individual cells from three independent experiments. Large points represent the means of individual experiments (n = 60 cells per experiment). Bars represent the mean ± SD from the three experiments. Significance was determined from biological replicates using a one-way ANOVA with Tukey's multiple comparisons tests. ns = not significant, *** p = 0.0009.

Figure R6 (new data; Figure EV12). Calcium chelation prevents ALG-2 recruitment to damaged lysosomes. (A) Confocal images of HeLa WT cells treated with or without BAPTA-AM followed by 1 mM LLOMe for 30 minutes and immunostained for ALG-2 and ALIX. Scale bars = 10 μ m for whole cell images and 2 μ m for insets. **(B)** Quantification of ALG-2/ALIX cell area from (A). Small points represent individual cells from three independent experiments. Large points represent the means of individual experiments ($n = 60$ cells per experiment). Bars represent the mean \pm SD from the three experiments. Significance was determined from biological replicates using a one-way ANOVA with Tukey's multiple comparisons tests. ns = not significant, **** $p < 0.0001$.

14. Minor points: Page 6, line 184: Specify the cell lines used. Line 188: Define the mutant ATG5.

Re14: This information has been added to the text

15. The recruitment of the ALG-2 LIR mutant to damaged lysosomes should be addressed.

Re15: To assess recruitment of the ALG-2 LIR mutant to damaged lysosomes, HA-ALG2^{WT} or HA-ALG2^{F32A} were stably expressed in HeLa cells. In agreement with the proposed hypothesis

that membrane ATG8ylation recruits ALG-2 to damaged membranes, the mutant failed to colocalize with the ESCRT machinery after LLOMe-induced damage (**Figure R7**). However, as described below in our response to Reviewer #2, Re3, we have opted to remove the LC3B-ALG2 interaction data from the revised manuscript.

Figure for referee with unpublished data and its description has been removed upon request by the authors.

16. Minor points: Several cropped images lack scale bars, and the scale value is not mentioned in the figure legends. The line used to measure intensity for histograms should also be shown.

Re16: Scale bars have been added throughout and yellow lines added to illustrate the position of intensity profiles.

17. The knockout of several proteins was not validated in this study, and either references or validation should be presented.

Re17: Additional western blot data (Figure EV5, Figure 6A) has been added where possible and references for all cell lines generated elsewhere have been added.

Referee #2:

The ESCRT machinery is crucial for repairing damaged endolysosomal membranes. However, how cells detect this damage and recruit ESCRT is not well understood. The current study by Corkery et al reveals that the translocation of autophagy-related ATG8 E3-like ligases to lysosomal membranes upon damage acts as a catalyst for ESCRT recruitment. The authors propose that leakage of protons or calcium from lysosomes triggers this recruitment through different pathways: V-ATPase-dependent for protons and sphingomyelin-dependent for calcium. The ATG16L1-ATG5-ATG12 or TECPR1-ATG5-ATG12 E3-like complexes are necessary for this process. The ATG5-ATG12 complex stabilizes the calcium sensor ALG-2 and recruits the repair complex, with LC3B binding directly to Ca²⁺-activated ALG-2. Thus, ATG8 E3-like ligases serve as damage sensors for ESCRT-mediated membrane repair.

Overall this is an interesting manuscript putting forward a new hypothesis on how lysosomal membrane damage is linked to ESCRT-III recruitment, an open question in the field. However, there are notable weaknesses in the proposed mechanism. In particular, the authors build on a model where ESCRT-III recruitment during lysosomal damage is thought to be mediated by the calcium-binding protein, ALG-2. However, this model is increasingly recognized to be damage type-specific, and likely does not apply to the specific perturbation applied here (LLOMe). A key piece of data is missing, where the requirement for ALG2 and its interaction with LC3 in ESCRT-III recruitment is functionally tested (see point 3 below).

Main concerns:

1- With regards to uniform vs fragmented distribution of ESCRT-III components upon ATG5-KO and ATG8-KO (Fig. 3C, 3D) - what is the distribution of ATG5 and ATG8 on lysosomes upon LLOMe treatment? Is it also uniform or are there certain locations where these proteins localize more? Does the uniformity of the distribution change upon single KO of TECPR1 or ATG16L1?

Re1: Due to the small size of lysosomes, imaging at a resolution sufficient to characterize protein architecture on damaged membranes is technically challenging and often limited by antibody quality. Therefore, to address the questions raised by the reviewer, we used expression of a constitutively active Rab5 mutant (Rab5^{Q79L}) to generate oversized early/late endosome-like vesicles. We confirmed that the oversized vesicles were acidified (**Figure R8A, new Figure EV8A**) and sensitive to damage by LLOMe (**Figure R8B, new Figure EV8B**) – thus providing a platform to assess the architecture of membrane repair proteins using traditional confocal microscopy.

Using this model, we shown that the ATG8 E3-like ligases, ATG8, ALG-2 and the ESCRT machinery all display a largely uniform distribution on damaged membranes in WT cells. The one exception is ATG8 which appears to show some localized enrichment (**Figure R9, new Figure EV8D**).

We further use this model to show that single KO of TECPR1 or ATG16L1 does not change the uniformity of ESCRT distribution (**Figure R10, new Figure EV9**).

Figure R8 (new data; Figure EV8A and B). Oversized Rab5^{Q79L}-induced vesicles are sensitive to damage by LLOMe. (A) Confocal images of a HeLa cell stably expressing Rab5^{Q79L}, stained with LysoTrackerRED. Nuclei were stained with Hoechst. Scale bar = 10 μ m. (B) Live cell imaging of HeLa cells stable expressing EGFP-Rab5^{Q79L}, co-transfected with mCherry-TECPR1, and treated with 0.5 mM LLOMe for 10 minutes. Scale bars = 10 μ m for whole cell images and 2 μ m for single vesicle insets.

Figure R9 (new data; Figure EV8D). Repair machinery localization on damaged oversized vesicles. Confocal images of oversized vesicles from HeLa cells stably expressing Rab5^{Q79L} treated with 0.5 mM LLOMe for 10 minutes. Scale bars = 2 μ m. Corresponding fluorescence intensity profiles are shown below (location marked by a yellow line on the merged image).

Figure R10 (new data; Figure EV9). Single knockout of ATG16L1 or TECPR1 does not impact ESCRT architecture at damaged membranes. (A) Confocal images of oversized vesicles from HeLa ATG16L1KO and TECPR1KO cells stably expressing EGFP-Rab5Q79L treated with 0.5 mM LLOMe for 10 minutes. Scale bars = 2 μ m. (B) Quantification of IST1 lysosomal area from (A). Grey points represent individual vesicles from three independent experiments ($n \geq 30$ vesicles per experiment). Significance was determined from biological replicates using a Student's t test. ns = not significant.

2- In the LysoTrackerRED LLOMe-washout experiment (Fig. 3E, 3F), how do the authors differentiate membrane repair with lysosome biogenesis or other vesicle trafficking processes? A better experimental set-up is to pre-load lysosomes with AF488 and AF568-labeled Dextran and repeat the same experimental paradigm to better differentiate repair of pre-existing lysosomes with lysosome biogenesis. This is an important distinction since ATG8-KO may also impair lysophagy processes.

Re2: Due to technical challenges regarding variable/inefficient dextran loading in the cell lines used in this study, we instead assessed susceptibility to lysosomal rupture using Gal3 staining as a marker of ruptured lysosomes (PMID: 38781206). In this experimental setup, cells are treated briefly with LLOMe to induce nanoscale damage to lysosomal membranes (ESCRT recruitment) without inducing rupture (Gal3 recruitment) (Figure R1, new Figure EV2). LLOMe was then washed off and the cells allowed to recover for 20 minutes, followed by immunostaining for ALIX and Gal3. The appearance of Gal3 staining following the removal of LLOMe signals progression from nanoscale damage to lysosomal rupture due to an inability to repair the damage, thus removing lysosome biogenesis from the equation. In this assay, we observed increased Gal3 staining in E3-DKO and ATG5KO cells, consistent with impaired ESCRT recruitment. ATG8KO cells, however, did not display increased Gal3 staining suggesting the intermediate ESCRT recruitment observed in this cell line is sufficient to provide protection against rupture (Figure R11, new Figure 4A). Combined with the LysoTracker re-acidification assay, this data suggests that in the absence of membrane ATG8ylation the ESCRT machinery can offer protection against damage expansion, but membrane repair and subsequent re-acidification requires membrane ATG8ylation.

Figure R11 (new data; Figure 4A). Assessing susceptibility to lysosomal rupture. (A) Schematic outline of lysosome rupture assay. Scale bars = 10 μm. (B) Quantification of ALIX and Gal3 area from (A). Small points represent individual cells from three independent experiments. Large points represent the means of individual experiments (n = 60 cells per experiment). Bars represent the mean ± SD from the three experiments. Significance was determined from biological replicates using a one-way ANOVA with Tukey's multiple comparisons tests. Comparisons are shown against the cell line-matched untreated sample. ns = not significant, * p = 0.0191, ** p = 0.0028, *** p = 0.0001, **** p < 0.0001..

3- Regarding recruitment of ALG-2 in an LIR-dependent manner to lysosomes - the authors are missing a few key data:

- Do ALG-2 LIR mutants fail to get recruited to damaged lysosomes (by IF)? Authors have only done pull-down experiments with these mutants, and only in the context of Ca²⁺ addition, but not LLOMe addition.

Re3: To assess recruitment of the ALG-2 LIR mutant to damaged lysosomes, we first attempted live-cell studies using the mCherry-ALG2 construct used in recent reports (PMID: 38781205). Unfortunately, mCherry-ALG2 is not recruited to lysosomes damaged by LLOMe (Figure R12) which we attribute to the presence of the bulky mCherry tag.

Figure for referee with unpublished data and its description has been removed upon request by the authors.

In an alternative approach, we stably expressed HA-ALG2^{WT} or HA-ALG2^{F32A} in ALG-2 KO cells and performed immunofluorescence in cells treated with or without LLOMe. In agreement with the

proposed hypothesis that membrane ATG8ylation recruits ALG-2 to damaged membranes, the mutant failed to colocalize with the ESCRT machinery after LLOMe-induced damage (**Figure R7**).

- Do ALG-2 LIR mutants fail to recruit ESCRT-III components upon lysosome damage by LLOMe? KD-rescue experiments are needed to substantiate this key point. Of note, in the Chen et al, 2024 (PMID: 38781205) manuscript, the authors found that ALG-2 dependent ESCRT-III recruitment was only required for GPN-dependent lysosome damage, but not LLOMe-dependent lysosome damage. Do the authors find the same in their system, and can they comment on the differences/similarities?

Re4: We thank the reviewer for this insightful comment as it has significantly reshaped the revised manuscript. To determine the dependence on ALG-2 for ESCRT recruitment to lysosomes damaged with LLOMe, we obtained HeLa ALG2 KO cells and performed immunofluorescence for IST1 in response to LLOMe treatment. In agreement with the 2022 study from the Mizushima lab (PMID: 35274304 – FigS3) we found ALG2 to be dispensable for ESCRT recruitment in response to LLOMe-induced damage (**Figure R13A-C, new Figure 6A-D**). However, we found that ALG2 KO cells have an increased susceptibility to lysosomal rupture (**Figure R13D, E, new Figure 6E, F**), suggesting ALG-2 plays a functional role in the repair process.

In light of this new data, we have opted to remove the LC3B-ALG2 interaction data from the revised manuscript. While we hypothesize that this interaction may be important for the membrane repair process, it is clear that it is not driving ESCRT recruitment and thus, outside the scope of this manuscript. Ongoing studies in the lab aim to explore how the architecture and functionality of the ESCRT machinery is regulated by ATG8-specific protein-protein interactions – a more appropriate platform for this data.

The dispensability of ALG-2 did, however, lead to a detailed exploration into the role of calcium in the ESCRT recruitment process, as calcium-dependence is often attributed to direct interaction between Ca^{2+} -activated ALG-2 and components of the ESCRT complex (see Reviewer#1, Re 13).

With respect to the differences observed between LLOMe and GPN, it seems likely that assembly of the ESCRT machinery differs depending on the type of stress applied. Simply considering that mCherry-tagged ALG-2 is recruited to GPN-damaged membranes (PMID: 38781205), but not to LLOMe-damaged membranes (**Figure R12**) alludes to fundamental differences in ESCRT architecture in response to these two stressors. It is important to note that GPN treatment, unlike LLOMe, has the additional consequence of releasing calcium from the ER (PMID: 30617110), which is sufficient to induce ALG-2 recruitment to ER exit sites (**Figure R14**) (PMID: 17196169, 38386713). Thus, caution must be taken to separate these two responses when using GPN to study ALG-2 function at damaged lysosomes.

Figure R13 (new data; Figure 6 ALG-2 is dispensable for ESCRT recruitment to damaged membranes, but required for efficient repair. (A) Western blot analysis of HeLa WT and ALG2 KO cells treated with 0.5 mM LLOMe for 15 minutes. (B) Representative confocal images of HeLa WT and ALG2 KO cells treated with or without 0.5 mM LLOMe for 15 minutes and immunostained for IST1 and LAMP1. Scale bars = 10 μ m. (C) Quantification of IST1 cell area from (B). Small points represent individual cells from three independent experiments. Large points represent the means of individual experiments ($n \geq 45$ cells per experiment). Bars represent the mean \pm SD from the three experiments. Significance was determined from biological replicates using a one-way ANOVA with Tukey's multiple comparisons tests. ns = not significant, *** $p = 0.0002$, **** $p < 0.0001$. (D) Quantification of IST1/LAMP1 colocalization from (B). Small points represent individual cells from three independent experiments. Large points represent the means of individual experiments ($n = 45$ cells per experiment). Bars represent the mean \pm SD from the three experiments. Significance was determined from biological replicates using a Student's t test. ns = not significant. (E) Representative confocal images of HeLa WT and ALG2 KO cells treated with or without 0.25 mM LLOMe for 5 minutes and immunostained for Gal3. Scale bars = 10 μ m. (F) Quantification of Gal3 puncta from (B). Small points represent individual cells from three independent experiments. Large points represent the means of individual experiments ($n \geq 72$ cells per experiment). Bars represent the mean \pm SD from the three experiments. Significance was determined from biological replicates using a one-way ANOVA with Tukey's multiple comparisons tests. ns = not significant, **** $p < 0.0001$.

Figure for referee with unpublished data and its description has been removed upon request by the authors.

Additional concerns on specific data:

- Figure 1B: It would be clearer if the authors show statistical comparisons between marker between genotype, not between marker within a genotype- as the conclusion being made is that the E3 DKO has less ALIX recruitment compared to WT but not the single KOs.

Re5: Figure 1B has been adjusted accordingly.

- Figure 1B: Why does it appear that the E3 DKO is not more damaged based on Gal3 recruitment? Have the authors done a time course to see whether the E3 DKO recruits Gal3 more quickly or accumulates more Gal3 over time?

Re6: This point has been addressed in response to Reviewer#1, Re3.

The reviewer's point regarding basal levels of damage is a good one. If we use Gal3 recruitment as an indicator of membrane rupture, we would anticipate that E3-DKO cells are more susceptible to damage and, therefore, progress to rupture more quickly. A 30-minute treatment with 1 mM LLOMe is quite harsh and would be sufficient to severely damage the whole lysosome population, so changes in dynamics would not be obvious under the conditions used for these experiments. To address this point, we performed a time course experiment to assess ALIX and Gal3 recruitment in WT and E3-DKO cells treated with 250 μ M LLOMe (**Figure R1, new Figure EV2**). Untreated cells do not show any basal levels of membrane damage (ALIX/Gal3) likely due to parallel ESCRT-independent membrane repair pathways capable of repairing endogenous damage (PMID: 35388011, 38348092, 36071159). At later time points (15/20 minutes) we do observe a significant increase in Gal3 staining in E3-DKO cells, relative to WT. This data suggests that the inability to recruit ESCRT machinery increases lysosomal susceptibility to rupture.

-Figure 3B: Is it that less ESCRT-III is recruited upon ATG8 KO or just that it is recruited slower?

Re7: With the additional data added using the Rab5^{Q79L}-induced oversized vesicles (**Figure 3D and E** in the revised manuscript), we believe that it is less ESCRT recruitment in ATG8KO cells, not slower recruitment kinetics.

-Figure 4A: The authors should quantify ALG-2 on lysosomes, as the differences will be clearer and more convincing.

Re8: Colocalization data between ALG-2 and LAMP1 has been added (**Figure R4, new Figure 5B**).

- In Figure 5A, the differences do not appear to be drastic by eye. The authors should do further in cell experiments: in LC3 WT or F52A expressing cells +/- LLOMe, whether ALG-2 recruitment is impaired, and likewise verify ALG-2 WT or F32A mutant recruitment +/- LLOMe.

Re9: See Re4 and Figure R7. The LC3B-ALG2 interaction data has been removed from the revised manuscript.

Dear Yaowen,

Thank you for submitting your point-by-point response. I forwarded it to both referees; at this stage though, one was not available to contribute any comments. After reading the feedback from the other referee and discussing your very careful point-by-point response with the editorial team here at EMBO Journal, I have concluded that all of your points are indeed addressable within a realistic time-frame. I would therefore be open to re-sending a revised version of the manuscript to the referees.

At this stage, however, I must stress that this revised manuscript must gain strong support from both referees for us to be able to move forwards towards publication.

I should add that it is EMBO Journal policy to allow only a single round of revision, so the evaluation of your manuscript may only consider this revised version.

We generally allow three months as standard revision time. As a matter of policy, competing manuscripts published during this period will not negatively affect our assessment of the conceptual advance presented by your study. Please contact me as soon as possible upon publication of any related work, to discuss how to proceed. If you envisage needing this three-month window to be widened, please let me know.

Thank you again for the opportunity to consider your work for publication. I look forward to reading your revision.

Best wishes,

William

William Teale, PhD
Editor
The EMBO Journal
w.teale@embojournal.org

When submitting your revised manuscript, please carefully review the instructions below and include the following items:

2) individual production quality figure files as .eps, .tif, .jpg (one file per figure).

3) a .docx formatted letter INCLUDING the reviewers' reports and your detailed point-by-point response to their comments. As part of the EMBO Press transparent editorial process, the point-by-point response is part of the Review Process File (RPF), which will be published alongside your paper.

4) a complete author checklist, which you can download from our author guidelines ([https://wol-prod-cdn.literatumonline.com/pb-assets/embo-site/Author Checklist%20-%20EMBO%20J-1561436015657.xlsx](https://wol-prod-cdn.literatumonline.com/pb-assets/embo-site/Author%20Checklist%20-%20EMBO%20J-1561436015657.xlsx)). Please insert information in the checklist that is also reflected in the manuscript. The completed author checklist will also be part of the RPF.

6) We require a 'Data Availability' section after the Materials and Methods. Before submitting your revision, primary datasets produced in this study need to be deposited in an appropriate public database, and the accession numbers and database listed under 'Data Availability'. Please remember to provide a reviewer password if the datasets are not yet public (see <https://www.embopress.org/page/journal/14602075/authorguide#datadeposition>). If no data deposition in external databases is needed for this paper, please then state in this section: This study includes no data deposited in external repositories. Note that the Data Availability Section is restricted to new primary data that are part of this study.

Note - All links should resolve to a page where the data can be accessed.

7) When assembling figures, please refer to our figure preparation guideline in order to ensure proper formatting and readability in print as well as on screen:
<http://bit.ly/EMBOPressFigurePreparationGuideline>

8) For data quantification: please specify the name of the statistical test used to generate error bars and P values, the number (n) of independent experiments (specify technical or biological replicates) underlying each data point and the test used to calculate p-values in each figure legend. The figure legends should contain a basic description of n, P and the test applied. Graphs must include a description of the bars and the error bars (s.d., s.e.m.).

9) We would also encourage you to include the source data for figure panels that show essential data. Numerical data can be provided as individual .xls or .csv files (including a tab describing the data). For 'blots' or microscopy, uncropped images should be submitted (using a zip archive or a single pdf per main figure if multiple images need to be supplied for one panel). Additional information on source data and instruction on how to label the files are available at .

10) We replaced Supplementary Information with Expanded View (EV) Figures and Tables that are collapsible/expandable online (see examples in <https://www.embopress.org/doi/10.15252/emboj.201695874>). A maximum of 5 EV Figures can be typeset. EV Figures should be cited as 'Figure EV1, Figure EV2' etc. in the text and their respective legends should be included in the main text after the legends of regular figures.

12) Our journal encourages inclusion of *data citations in the reference list* to directly cite datasets that were re-used and obtained from public databases. Data citations in the article text are distinct from normal bibliographical citations and should directly link to the database records from which the data can be accessed. In the main text, data citations are formatted as follows: "Data ref: Smith et al, 2001" or "Data ref: NCBI Sequence Read Archive PRJNA342805, 2017". In the Reference list, data citations must be labeled with "[DATASET]". A data reference must provide the database name, accession number/identifiers and a resolvable link to the landing page from which the data can be accessed at the end of the reference. Further instructions are available at .

13) In order to increase the reproducibility and reach of your work, The EMBO Journal includes a table of reagents that were used in the study. Please provide this along with your revisions.

Further instructions for preparing your revised manuscript:

We realize that it is difficult to revise to a specific deadline. In the interest of protecting the conceptual advance provided by the work, we recommend a revision within 3 months (14th Dec 2025). Please discuss the revision progress ahead of this time with the editor if you require more time to complete the revisions. Use the link below to submit your revision:

Same as the Appeal Letter.

Dear Yaowen,

Thank you for submitting the revised version of your manuscript, which addresses the concerns of the referees. This revised version has now been re-reviewed; I attach the second referee reports to the bottom of this mail. As you will see, you have addressed the referees' concerns to their satisfaction. Before I can finally accept the manuscript, there are some remaining editorial points which need to be addressed. In this regard, would you please:

- include up to five keywords,
- rename the Conflict of Interest section the "Disclosure Statement and Competing Interests" statement,
- remove the author credit section from the manuscript file,
- upload the synopsis image as a .jpeg or a .tif file,
- provide a two sentence statement and 3-5 bullet points that capture the key findings of the paper,
- define the annotated p values ****/****/*/* as well as provide the exact p-values for the same in the legend of figure 3E,
- provide exact p values in the legends of figures 1C, I; 2B, C; 3B,4B, D; 6C, F; EV5 B; S1 B, S4 C, S6 B,
- indicate the statistical test used for data analysis in the legend of figure 3B, and
- define the error bars in the legend of figure 3E.

During our routine image checks, we noticed that the blots across the figure set appear pixelated. This is a common result of converting original 16-bit TIFF images to RGB format for publication, and while not a cause for concern, it can sometimes give the impression of image alteration to critical readers.

To resolve this, I reviewed the high-resolution TIFF source data you provided for Figures 1-6 and can confirm that the original images are of high resolution without pixelation.

To maintain consistency and avoid any possible misinterpretation, I kindly ask that you upload the source data for Figures EV1-EV5. This will allow us to confirm the integrity of the full figure set and support transparency for readers.

Please also upload the Appendix file at a higher resolution using the original 16-bit TIFF images.

I look forward to receiving these changes. EMBO Press is an editorially independent publishing platform for the development of EMBO scientific publications.

Best wishes,

William

William Teale, PhD
Editor
The EMBO Journal
w.teale@embojournal.org

We realize that it is difficult to revise to a specific deadline. In the interest of protecting the conceptual advance provided by the work, we recommend a revision within 3 months (10th Feb 2026). Please discuss the revision progress ahead of this time with the editor if you require more time to complete the revisions. Use the link below to submit your revision:

Referee #1:

The authors addressed my main concerns.

Referee #2:

The authors have addressed my previous concerns satisfactorily. In particular, removing the claim of ALG2-dependent ESCRT recruitment adds clarity to the manuscript and focuses it on the key novelty point for the role of E3-like proteins as early damage sensors. I support acceptance and publication of the work.

All editorial and formatting issues were resolved by the authors.

Dear Yaowen,

I am pleased to inform you that your manuscript has been accepted for publication in the EMBO Journal.

Congratulations to you and all involved!

Best wishes,

William

William Teale, PhD
Editor
The EMBO Journal
w.teale@embojournal.org

Please note that it is The EMBO Journal policy for the transcript of the editorial process (containing referee reports and your response letters) to be published as an online supplement to each paper. If you should prefer removal of any referee-only figures included in the point-by-point response(s), e.g. because they may still be used for future publication or because they have been reproduced from published work by others, please do let us know immediately via response email.

More information is available here: https://www.embopress.org/transparent-process#Review_Process